

# ICGEM – 15 years of successful collection and distribution of global gravitational models, associated services and future plans

E. Sinem Ince[1], Franz Barthelmes[1], Sven Reißland[1], Kirsten Elger[2], Christoph Förste[1], Frank Flechtner[1,4], and Harald Schuh[3,4]

[1]Section 1.2: Global Geomonitoring and Gravity Field, GFZ German Research Centre for Geosciences, Potsdam, Germany
[2]Library and Information Services, GFZ German Research Centre for Geosciences, Potsdam, Germany
[3]Section 1.1: Space Geodetic Techniques, GFZ German Research Centre for Geosciences, Potsdam, Germany
[4]Department of Geodesy and Geoinformation Science, Technical University of Berlin, Berlin, Germany

*Correspondence to*: E. Sinem Ince (elmas.sinem.ince@gfz-potsdam.de)

**Abstract.** The International Centre for Global Earth Models (ICGEM, http://icgem.gfz-potsdam.de/) hosted at the GFZ German Research Centre for Geosciences (GFZ) is one of the five Services coordinated by the International Gravity Field Service (IGFS) of the International Association of Geodesy (IAG). The goal of the ICGEM Service is to provide the scientific community with a state of the art archive of static and temporal global gravity field models of the Earth, and develop and operate interactive calculation and visualisation services of gravity field functionals on user defined grids or at a list of particular points via its website. ICGEM offers the largest collection of global gravity field models, including those from the 1960s, as well as the most recent ones that have been developed using data from dedicated gravity missions, advanced processing methodologies and additional data sources such as satellite and terrestrial gravity. The global gravity field models have been collected from different institutions at international level and after a validation process made publicly available in a standardized format with DOI numbers assigned through GFZ Data Services. The development and maintenance of such a unique platform is crucial for the scientific community in geodesy, geophysics, oceanography, and climate research. The services of ICGEM have motivated researchers worldwide to grant access to their gravity field models and also provide them an access to variety of other gravity field models and their products. In this article, we present the development history and future plans of ICGEM and its current products and essential services. We present the Earth's static, temporal, and topographic gravity field models as well as the gravity field models of other celestial bodies together with examples produced by the ICGEM's calculation and 3D visualisation services and give an insight how the ICGEM Service can additionally contribute to the needs of research and society.





## 1 Introduction

The determination of the Earth's gravity field is one of the main tasks of geodesy. With the highly accurate satellite measurements as result of today's advancing technology, it is now possible to represent the Earth's global gravity field and its variations with better spatial and temporal resolutions compared to the first generation global gravity field models derived in the 1960s. Global gravity field models provide information about the Earth's shape, its interior and fluid envelope and mass variations which give hints to climate related changes in the Earth system. The computation of gravity field functionals (e.g. geoid undulations, gravity anomalies) from the model representation is therefore not only relevant for geodesy, but also for other geosciences, such as geophysics, glaciology, hydrology, oceanography, and climatology.

Some application examples in which the precise knowledge of the Earth's gravity field is fundamental are: (1) To establish a global vertical datum of global reference systems (Sideris and Fotopoulos, 2012), (2) to monitor mass distributions that are indicators of climate related changes (Tapley et al., 2004, Schmidt et al., 2006), (3) to simulate the perturbing forces on space vehicles and predict orbits in aeronautics and astronautics (Chao, 2005), (4) to explore the interior structure and geological evolution of our Earth (Wieczorek, 2015), and (5) to explore minerals or fossil fuels and to examine geophysical models developed using gravity inversion (Oldenburg et al., 1998; Bosch and McGaugney, 2001). For most of the above mentioned examples, representation of the Earth's global gravity field in terms of mathematical models is an indispensable need. For such models having plenty of vital applications, it is necessary to develop strategies for: (1) Using the most recent datasets and technologies in the field of gravity field determination (Floberghagen et al., 2011; Flechtner et al., 2014), (2) processing the raw data in different forms and making the validated models publicly available in citable form (Barthelmes, 2014; Drewes, 2016), and (3) developing sophisticated calculation and visualisation tools that are useful for both experts, young scientists, students, and for the general public (Barthelmes, 2014; Drewes, 2016; Toth, 2018).

There are various complementary data resources used for the development of high quality global gravity field models. For example, advanced satellite observations are one of them and they can be in the form of satellite orbital perturbations derived from GNSS measurements, microwave/laser range rate measurements between two satellites, or satellite laser ranging (SLR) observations from the Earth's surface to the near Earth satellites. Recent satellites contributing to the tremendous improvements in global gravity field modelling are the dedicated gravity missions CHAMP (Reigber et al., 2002), GRACE (Tapley et al., 2004), GRACE Follow-On (Flechtner et al., 2014; Flechtner et al., 2016), GOCE (Drinkwater et al., 2003); and SLR satellites such as LAGEOS 1 and LAGEOS 2; as well as the fleet of altimetry satellites such as Topex/Poseison and Jason 1 and 2. Other fundamental datasets used in the development of global gravity field models are non-gravitational accelerations and gravity gradients measured on-board spacecraft, terrestrial gravity measurements collected on moving platforms (e.g. airborne and shipborne) and finally actual terrestrial gravity measurements collected on the Earth's surface. Besides the gravity measurements, high resolution digital elevation models (DEMs) complement the global gravity field models for mapping detailed features of the gravity field and in the areas with missing real gravity measurements such as Antarctica.



Static and temporal global gravity field models are developed based on different mathematical approaches. These approaches are designed to take the advantage of each of the above mentioned measurement technique with the overarching goal of mapping the Earth's gravity field with its smallest details possible and monitor its temporal variations. Different institutions and agencies study and improve these techniques and develop gravity field models for different applications and regular
updates when new measurements become available from satellites and terrestrial measurements. The International Centre for Global Earth Models (ICGEM) contributes to the collection and validation of these models to make them freely available online, and provides interactive calculation and visualisation services. Therefore, it has naturally become the meeting point for both the model developers and the users of the global gravity field models.

ICGEM is one of the five Services coordinated by the International Gravity Field Service (IGFS) (http://igfs.topo.auth.gr) of
10 the International Association of Geodesy (IAG, http://www.iag-aig.org). The IAG is the global scientific organization in the field of geodesy which promotes scientific cooperation and research in geodesy and contributes to it through its various research bodies. The roots of the IAG can be traced back to the 19th century. Today it is one of the largest organizations in geodetic and geophysical research, especially thanks to the extensive services it provides. The IAG is a member of the International Union of Geodesy and Geophysics (IUGG, http://www.iugg.org) which itself is a member of the International
Science Council (ISC, https://council.science). Within the same hierarchy, the IGFS as an IAG Service is a unified "umbrella", which: (1) coordinates the collection, validation, archiving and dissemination of gravity field related data, (2) coordinates courses, information materials and outreach to the general public outreach related to the Earth's gravity field, (3) unifies gravity products for the needs of the Global Geodetic Observing System (GGOS, www.ggos.org). The Services of the IGFS and corresponding responsible institutions are shown in Figure 1.

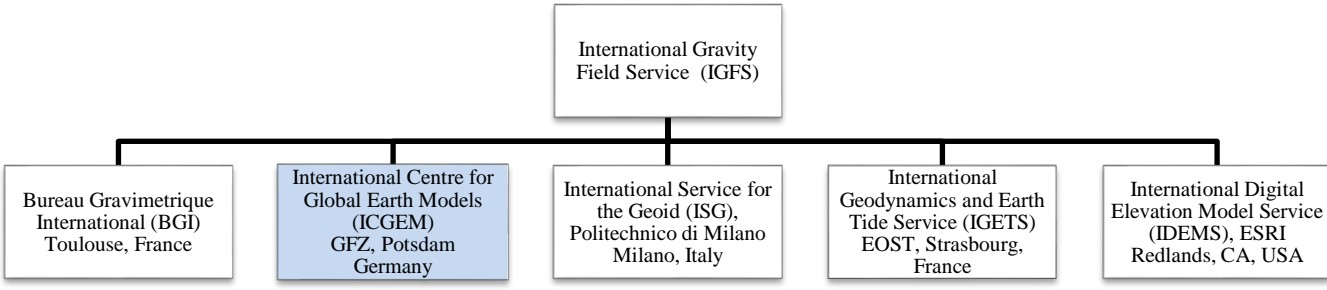

**Figure 1: The IGFS Services and hosting institutions.**

The 15 year old ICGEM Service has been collecting and archiving almost all of the existing static global gravity field models available worldwide. During the last few years, due to requests of users and model developers, ICGEM started to collect also temporal gravity field models and provide links to the original contributors. Since its establishment, ICGEM has structured
itself based on users' needs and nowadays provides the following services:



- Collecting and long term archiving of existing static global gravity field models, solutions from dedicated shorter time periods (e.g. monthly GRACE models), and recently topographic gravity field models, and making them available on the web in a standardized format (Barthelmes and Förste, 2011),

- Since late 2015, the above service has been extended with the possibility of assigning Digital Object Identifiers (DOI) to the models, i.e. to the datasets of coefficients,
- A web interface to calculate gravity field functionals from the spherical harmonic models on freely selectable grids and user defined point coordinates,
- A 3D interactive visualisation service for the gravity field functionals (geoid undulations and gravity anomalies) using static and time variable gravity field models (Drewes, 2016; Toth, 2018),

- Quality checks of the models via comparisons with other models in the spectral domain and also with respect to GNSS/levelling derived geoid undulations at benchmark points collected for different countries,
- The visualisation of surface spherical harmonics as tutorial,
- The theory and formulas of the calculation service documented in GFZ's Scientific Technical Report STR09/02 (Barthelmes, 2013),

- Manuals and tutorials for global gravity field modelling and usage of the service (Barthelmes, 2014), and
- The ICGEM web-based gravity field discussion forum.

The tasks of the other four Services are the following (http://www.iag-aig.org/index.php?tpl=cat&id_c=11):

- The Bureau Gravimetrique International (BGI) collects all terrestrial measurements and pertinent information worldwide related to the Earth gravity field; compiles, stores, and redistributes them based on requests.

- The International Service for the Geoid (ISG) collects worldwide geoid data, conducts research and collects and distributes software for geoid determination, and organises Geoid schools.
- The International Geodynamics and Earth Tide Systems (IGETS) provides a service to monitor temporal variations of the Earth gravity field through long-term records from ground gravimeters and other geodynamic sensors via the database run by GFZ.

- Finally, the International Digital Elevation Model Service (IDEMS) distributes the data and information about Digital Elevation Models, relevant software and related datasets which are publicly available.

The five Services exchange information via the IGFS and collaborate in the future plans of geodetic and gravity field related activities, such as GGOS which aims to advance our understanding of the dynamic Earth system by quantifying the changes of our planet in space and time (www.ggos.org).

With this paper, we inform ICGEM's potential users about the content and the services that the ICGEM provides. We describe different global gravity field models archived in the Service, and provide examples of their use for different purposes via ICGEM's interactive calculation and 3D visualisation tools and give a summary of the documents available on the Service such as tutorials for undergraduate and graduate students. Finally, we share the future aspects of the Service that aims to bridge the future plans of the ICGEM Service with the potential users' needs.

The manuscript is organised as follows. We provide an overview of the general and scientific background of the ICGEM Service in Section 2 together with the details of different gravity field models provided by ICGEM. Details and examples of ICGEM's new features and various services such as Calculation and 3D Visualisation Services with examples are given in Section 3.





Documentation and details on the web programming of the new website which has been implemented in May 2016 are given in Section 4 and 5, respectively. Finally, in Section 6, we provide a summary and plans for future developments and aspects of the ICGEM Service. The sections are written independently which enables the reader to directly refer to the relevant section without reading the previous parts nor the entire paper.

## 2 The background of the ICGEM Service

### 2.1 History of ICGEM

In the second half of the 1990s, the demand for a single access point to the collection and distribution of gravity field models and associated services arose from an interdisciplinary scientific community that included geodesists, geophysicists, oceanographers and climate scientists. With the IGFS's initiation in 2003 and the hosting and financial support of GFZ, the ICGEM Service was established in the same year. The ICGEM Service was initially established to collect static global gravity field models under one umbrella and provide easy access to the models via its website without any required user registration. Different models developed based on different combination of datasets, serve for variety of different purposes. The old models are collected to be included in the archive, whereas the newer models are used in the modelling of the Earth's gravity field with its finest details and its temporal variations due to different reasons, e.g. mass redistributions due to climate change. The interest in the development and application of the static as well as the temporal gravity field models has increased significantly with the launch of the dedicated gravity field satellite missions such as CHAMP, GRACE, and GOCE. As a consequence, the ICGEM Service has become a unique platform for the largest and most complete collection of the static and temporal gravity field models.

The number of static gravity field models developed since the 1960s with respect to time is shown in Figure 2a. The launch of the gravity satellite missions stimulated the studies in global gravity field determination as indicated by the increased number of the models. The details of the features resolved by some of the selected satellite only models in the spatial domain are shown in Figure 2b. Each new satellite only model shows improvement due to the high quality data retrieval. For example, the uncertainties in the geoid signal have been reduced from tens of meters to ~10 cm, whereas the spatial details that can be resolved from the satellite only global models have been improved from thousands of km to about 120 km. It is important to recall that the CHAMP mission was a break through mission and increased the details provided by the global gravity field models drastically in the spatial domain from about 1500 to 300 km. One of the first models with CHAMP contribution has become famous as the "Potsdam Gravity Potato" (personal communication, Reigber and Schwintzer, 2002).





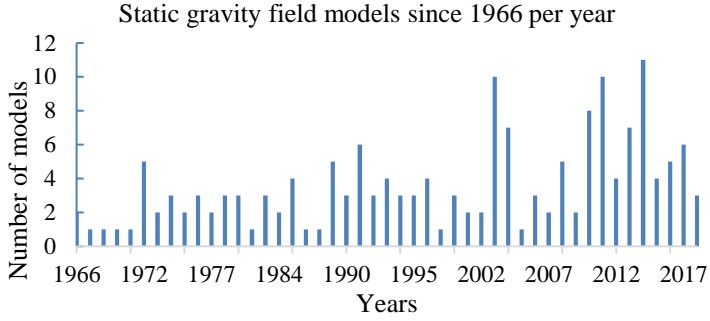

a)

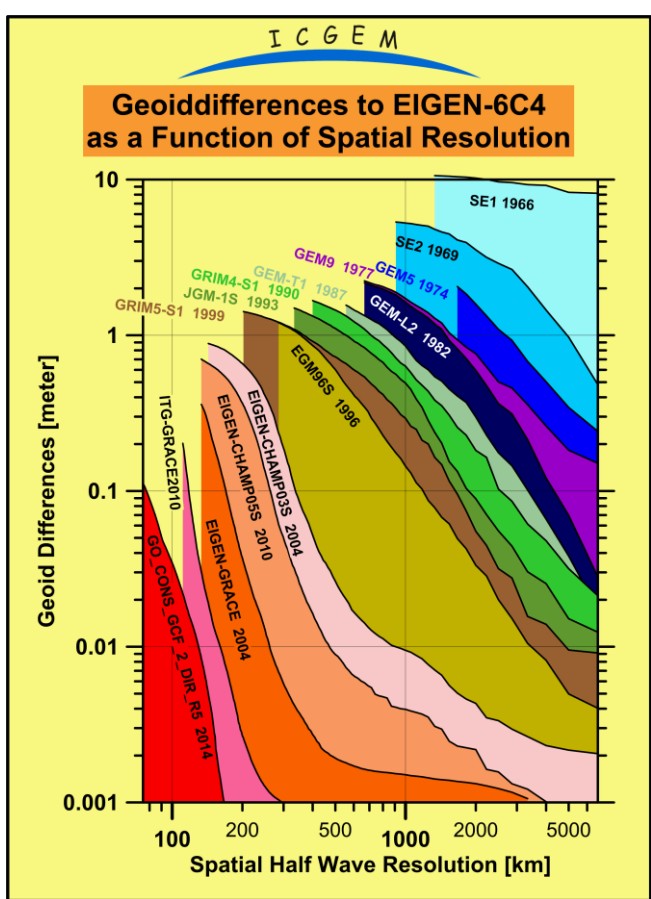

b)

**Figure 2: a) Number of the static gravity field models per year released since 1966. Note the increased number of models due to the launch of the dedicated satellite gravity missions after 2003 and 2011. b) The history of the improvement of the spatial resolution and the accuracy of the satellite only gravity field models. Signal amplitude difference over the years w.r.t. one of the latest combined global gravity field models, EIGEN-6C4 is shown (http://icgem.gfz-potsdam.de/History.png). Note that the EIGEN-6C4 is not the truth but a better approximation to the real gravity field. The uncertainties of the geoid signal have been reduced together with improved spatial resolution. See also the large improvement between the EGM96 and EIGEN-CHAMP03 due to the contribution of the CHAMP measurements.**





Currently, the ICGEM Service provides access to 168 static gravity field models (listed in Appendix 2), more than 20 temporal gravity field models (including different releases from same institutions between the years 2002 and 2016) (listed in Appendix 3), 18 topographic gravity field models (listed in Appendix 4) and finally models for three other celestial objects, 6 models for

Mars, 18 models for the Moon and 2 models for Venus.

With the recently launched GFZ and NASA mission GRACE-FO (Flechtner et al., 2014) and ongoing reprocessing efforts of GOCE (Siemes, 2018) and GRACE mission data (Dahle et al., 2018; Save et al., 2018; Yuan, 2018) we expect to receive new releases of better quality static and temporal gravity field models in the following years. Considering the number of visits of the ICGEM Website during the last few years, it has become obvious that the ICGEM Service has been recognised as a highly

demanded service by the community and used very actively worldwide and promises future developments. The distribution of the ICGEM service users by continents and the corresponding numbers between May 2018 and December 2018 are shown in Figure 3.

The ICGEM plans to continue its long term services in the future with the contributions from the reprocessed GOCE and GRACE data, the GRACE-FO mission as well as New Generation Gravity Missions. Moreover, new services such as provision

of time series of the changes of the gravity field of the Earth due to the flattening retrieved from SLR measurements from different institutions and agencies are some of our future plans. Accordingly, a new modernised and more flexible ICGEM website for future developments was necessary which indeed was realised and made available by GFZ in May 2018. A scheme of the current website structure of the ICGEM Service is presented in Figure 4. The list of the models, calculation and visualisation services and other services are accessible via the menu listed on the homepage (http://icgem.gfz-potsdam.de) as

shown in Figure 4.

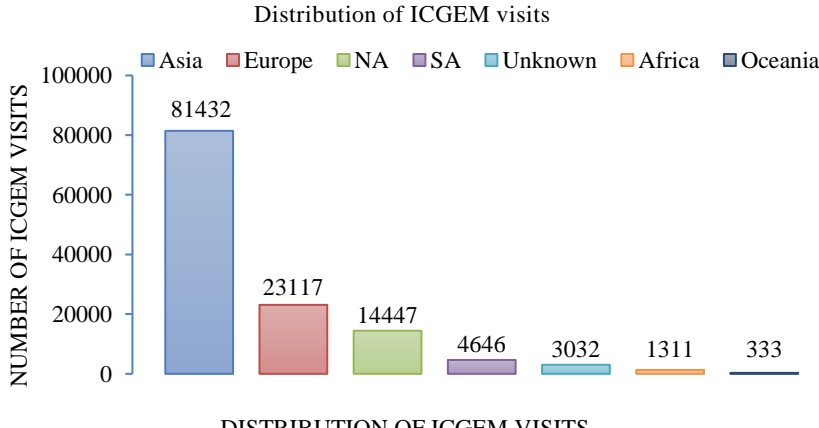

**Figure 3: Distribution of ICGEM use over continents between May 2018 and December 2018. NA is for North America, SA is for South America and Unknown is for anonymous entries.**






**Figure 4: The homepage structure of the new ICGEM service website.**



## 2.2 Scientific background and ICGEM's Data

The global gravity field model of the Earth is a mathematical model which describes the potential of the gravity field of the Earth in the 3 dimensional space. The terms gravity field and gravitational field are commonly mixed or used interchangeably. Therefore, we will start with the basic definition of the two terms for the sake of clarity in the rest of the paper. In geodesy,

there is a clear difference between the two terms. Gravitational potential, $V$ is formed by the summation of pure attractive forces of the masses of the Earth. Gravity field potential, $W$ is the sum of the potential of the Earth's gravitational attraction and potential of the centrifugal force acting on the masses rotating together with the Earth including on the ground and in the atmosphere.

Within this concept, a global model of the Earth's gravity field is a mathematical function which approximates the real gravity

field potential and allows to compute physical quantities related to the gravity field, i.e. the gravity field functionals at any position in the 3 dimensional space. A gravity field model should therefore contain both, a model of the gravitational potential and a model of the centrifugal potential. Because the modelling of the centrifugal potential is well known and can be done very accurately (Hofmann-Wellenhof and Moritz, 2006), the relevant and challenging part of a gravity field model is the modelling of the gravitational field. Therefore, the term gravity field model is also very often used in the sense of gravitational field

model. Through this article, we use the term "gravity field" since it has been commonly used in the past by the "gravity field" community.

Due to mass redistribution on, inside and outside the Earth caused by different reasons, the gravity field changes with respect to time. Although these temporal changes are very small and/or very slow, they can be measured (e.g. using GRACE mission data) and modelled up to a certain spatial and temporal resolution (Wahr et al., 1998; Wahr et al., 2004; Schmidt et al., 2006).

ICGEM provides access also to these temporal gravity field models.

According to Newton's law of gravitation, two point masses $m_1$ and $m_2$ separated by a distance of $l$, attract each other with a force:

$$F = G \frac{m_1 m_2}{l^2},$$    **Eq. (1)**

where $G$ is the gravitational constant. Although $m_1$ and $m_2$ attract each other symmetrically, it is convenient to call one of

them attracting mass and the other one attracted mass. For simplicity, setting the attracted mass equal to unity, the formulation is transformed into:

$$\vec{F} = G \frac{m}{l^2} \frac{\vec{l}}{l},$$    **Eq. (2)**



$m$ being the attracting mass and the force indicated as a vector. It is the expression of the force applied by the attracting mass $m$ onto a unit mass at a point distanced by $l$. Here the potential of gravitation for the two mass points can be introduced, which is a scalar function:

$$V = G\frac{m}{l},$$    **Eq. (3)**

with $\nabla V = \vec{F}$ where the $\nabla$ is the nabla operator and $\nabla V$ is the gradient of the gravitational potential.

The gravitational potential of a system consisting of infinite number of infinitely small volume elements $dv(x, y, z)$ with density $\rho(x, y, z)$, can be represented by:

$$V(x, y, z) = G\iiint_v \frac{\rho(x', y', z')}{l^2}dv,$$    **Eq. (4)**

where the integral is computed over the entire body, $\rho dv$ is the mass element and $l = \sqrt{(x-x')^2 + (y-y')^2 + (z-z')^2}$.

The gravitational potential satisfies the Poisson's equation:

$$\nabla^2 V = -4\pi G\rho,$$    **Eq. (5)**

where $\nabla^2$ is called the Laplace operator which is a differential operator given by the divergence of the gradient of $\nabla$ (Heiskanen and Moritz, 1967). Geodesy describes the gravitational potential only in empty space, outside the masses, and estimates a mathematical function that approximates the potential. Outside the masses, where the density $\rho$ is zero, the

gravitational potential satisfies the Laplace condition:

$$\nabla^2 V = 0.$$    **Eq. (6)**

With this condition satisfied, $V$ is a harmonic function in empty space.

Models approximating the real gravity field can be developed based on different mathematical representations, e.g. ellipsoidal harmonics, spherical radial basis functions, or spherical harmonic wavelets which are all harmonic outside the masses (outer

Earth). In practice, solid spherical harmonics are the ones widely used to represent the gravitational potential (or geopotential) globally. Solid spherical harmonics are an orthogonal set of solutions to the Laplace equation represented in a system of spherical coordinates (Heiskanen and Moritz, 1969; Hofmann-Wellenhof & Moritz, 2006).

The datasets available via the ICGEM Service are the spherical harmonic coefficients, which together with the spherical harmonic functions, approximate the real gravitational potential of the Earth and/or its variations. The spherical harmonic (or

Stokes') coefficients represent the global structure and irregularities of the geopotential field in the spectral domain (Heiskanen



and Moritz, 1967; Moritz, 1980; Hofmann-Wellenhof and Moritz, 2006; Barthelmes, 2013) and the formulation of the relationship between the spatial and spectral domain of the geopotential is expressed as:

$$V(r,\varphi,\lambda) = \frac{GM}{r} \sum_{l=0}^{l_{max}} \sum_{m=0}^{l} \left( \frac{R}{r} \right)^{l} \bar{P}_{lm}(\sin\varphi) \left( \bar{C}_{lm} \cos m\lambda + \bar{S}_{nm} \sin m\lambda \right), \qquad \textbf{Eq. (7)}$$

where $V$ is the Gravitational potential, $r,\varphi,\lambda$ correspond to the spherical geocentric coordinates of computation point

(radius, latitude and longitude, respectively),

$R$   is a (mathematically arbitrary) reference radius (in geodesy usually the mean semi-major axis of Earth is used),

$GM$   is the Gravitational constant times the mass of the Earth,

$l,m$   are degree and order of spherical harmonic, respectively,

$l_{max}$   is the maximum degree (and order) of the model expansion,

$\bar{P}_{lm,}$   are fully normalized Legendre polynomials of degree $l$ and order $m$,

$\bar{C}_{lm,}\bar{S}_{lm,}$ are fully normalized Stokes' coefficients.

Spherical harmonics are calculated using spherical coordinates and the normalisation represents when the average square value of the normalised harmonics integrated over the sphere is equal to unity (Heiskanen and Moritz, 1967) and it is represented by:

$$\frac{1}{4\pi} \int_{\lambda=0}^{2\pi} \int_{\varphi=-\pi/2}^{\pi/2} \left[ P_{lm} \sin\varphi \cos m\lambda \right]^{2} \cos\varphi\, d\varphi\, d\lambda = 1. \qquad \textbf{Eq. (8)}$$

The very low degree and order spherical harmonic functions can be physically defined and easily illustrated. For example, the $\bar{C}_{00}$ describes the mass of the Earth by scaling the value of $GM$, the whole mass of the Earth times the Newtonian constant. Therefore its value is close to 1. The degree 1 spherical harmonic coefficients, $\bar{C}_{10,}\bar{C}_{11,}$ and $\bar{S}_{10}$ are related to the coordinates of the geocentre and if the coordinate system's origin coincides with the geocentre, they are equal to zero. The coefficients

$\bar{C}_{21}$ and $\bar{S}_{21}$ are related to the mean rotational pole position. A tutorial on the representation of the spherical harmonics is available on the ICGEM website (http://icgem.gfz-potsdam.de/vis3d/tutorial) and an example of three different degree and order spherical harmonics are shown in Figure 5.

Mathematical representation of a gravity field model using summation of spherical harmonics is displayed in Figure 6. Sectorial, zonal, and tesseral spherical harmonic functions multiplied by the corresponding coefficient values are used to





develop gravity field model of the Earth expanded up to degree $l$ and order $m$ . The spherical harmonic degree expansion

corresponds approximately to the spatial resolution of $\lambda_{\deg ree} = \dfrac{180^{\circ}}{l}$ or $\lambda_{km} = \dfrac{20000 km}{l}$ , where 20000 km is the half

wavelength of the equatorial length and $l$ is the spherical harmonic degree. A spherical harmonic model of the gravity field

up to maximum degree $l_{max}$ consists of $\left(l_{max}+1\right)^{2}$ coefficients (see also Figure 6).

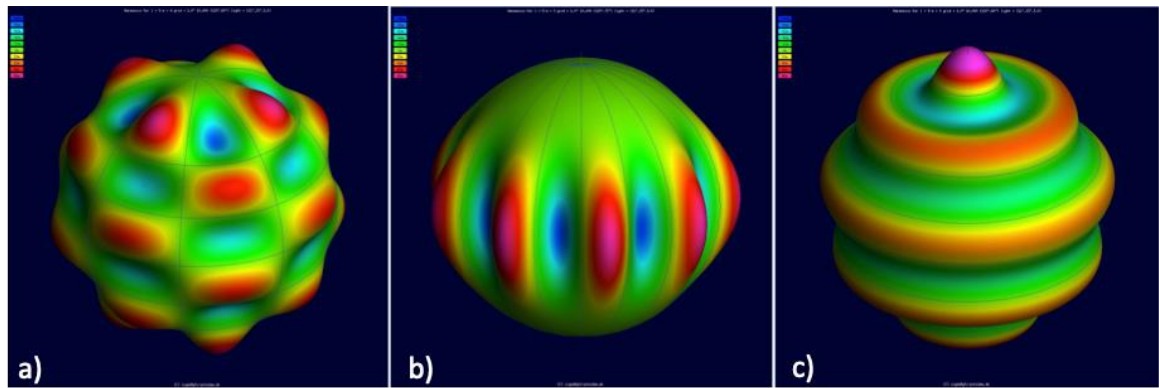

**Figure 5: 3D Visualisation of spherical harmonics as tutorial. The images show one specific surface spherical harmonic of degree $l$ and order $m$ such as a) tesseral ($l=9$, $m=4$) b) sectorial ($l=9$, $m=9$) c) zonal ($l=9$, $m=0$) spherical harmonics.**

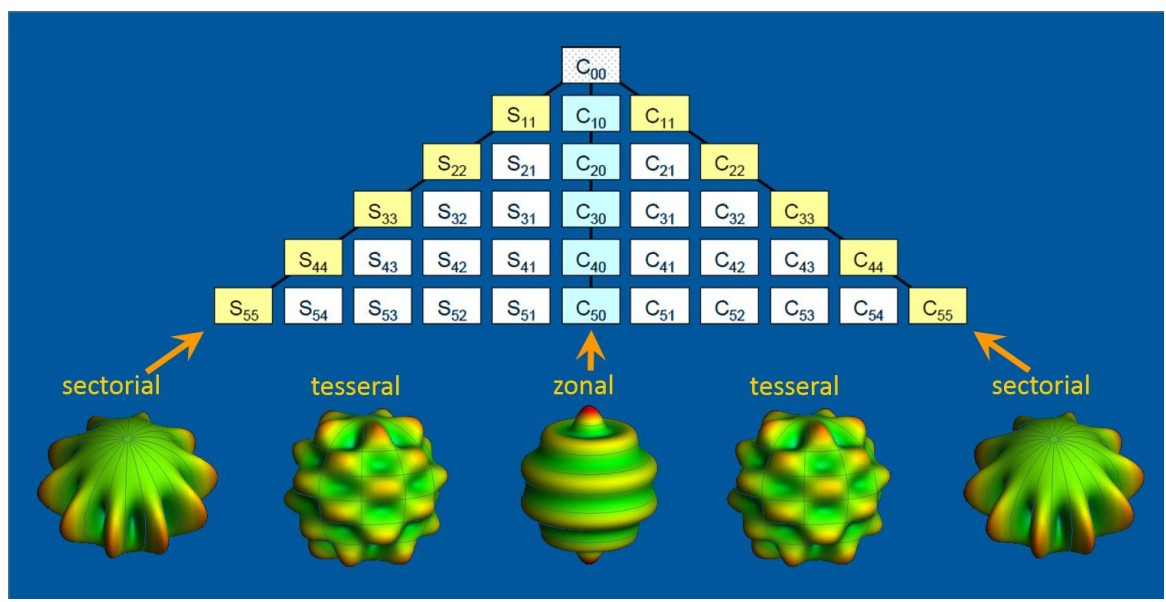

10 **Figure 6: Mathematical representation of gravity field potential using sectorial, tesseral and zonal spherical harmonics. A spherical approximation of the gravity field up to maximum degree of $l_{max}$ consists of $(l_{max}+1)^{2}$ coefficients (credit Barthelmes, F.).**



The terms $\bar{C}_{lm}$ and $\bar{S}_{lm,}$, and their variations are the fundamental data of the ICGEM Service that are retrieved from real gravity measurements and satellite observations as well as forward modelling using high resolution digital elevation models. Moreover, these coefficients are used in the calculation of the gravity field functionals directly.

At this point, it is worth mentioning about the mathematically defined normal potential which helps to approximate the real gravity potential for practical reasons. For many purposes, it is useful and sufficient to approximate the figure of the Earth by a reference ellipsoid. This is defined as the ellipsoid of revolution which fits the geoid, i.e. the undisturbed sea surface of the Earth and its fictitious continuations below the continents, as good as possible (i.e. in the sense of least squares fit). The normal potential together with the geometrical ellipsoid establish the Geodetic Reference System, e.g. WGS84 or GRS80 (Moritz,

1980; Mularie, 2000). Like the gravity potential $W$, the normal potential $U$ also consists of a gravitational potential and centrifugal potential. The attracting part of the normal potential can also be represented in terms of spherical harmonics. Due to the rotational symmetry, the expansion of ellipsoidal normal potential contains only the terms for $m = 0$ and degree $l =$ even. In most cases, the coefficients of $\bar{C}_{00}^{U}, \bar{C}_{20}^{U}, \bar{C}_{40}^{U}, \bar{C}_{60}^{U}$, and $\bar{C}_{80}^{U}$ are used in the calculation of the normal potential, where the superscript $U$ indicates the normal potential.

Using the normal potential, the real gravity field potential can be split into two parts, the normal potential and the disturbing potential as expressed below:

$$W(r, \varphi, \lambda) = U(r, \varphi) + T(r, \varphi, \lambda) .$$
**Eq. (9)**

If we subtract the Stokes' coefficients ($\bar{C}_{00}^{U}, \bar{C}_{20}^{U}, ..., \bar{C}_{80}^{U}$) of an ellipsoidal normal potential, $U(r, \varphi)$ from the gravity potential, the disturbing potential $T(r, \varphi, \lambda)$ can also be mathematically represented in terms of spherical harmonics,

$\bar{C}^{'} = \bar{C} - \bar{C}^{U}$ .

The disturbing potential is a 3d function and valid in space and it can be represented by:

$$T(r, \varphi, \lambda) = \frac{GM}{r} \left[ \sum_{l=0}^{l_{max}} \left( \frac{R}{r} \right)^{l} \sum_{m=0}^{l} \bar{P}_{lm}(\sin \varphi) \left( \bar{C}_{lm}^{'} \cos m\lambda + \bar{S}_{lm}^{'} \sin m\lambda \right) \right] .$$
**Eq. (10)**

The two fundamental gravity field functionals used in geosciences very often are geoid undulation and gravity disturbances which in practice can be approximately calculated using the disturbing potential. It is worth recalling that the geoid undulation

is the distance between the particular equipotential surface (geoidal surface) and the surface of the reference ellipsoid (conventional ellipsoid of revolution). The gravity disturbance on the other hand is the difference of the magnitude of the





gradient of the Earth's potential (the gravity) and the magnitude of the gradient of the normal potential (the normal gravity) at the same point (e.g. Earth's surface), $\delta g(r,\varphi,\lambda) = |\nabla W(r,\varphi,\lambda)| - |\nabla U(r,\varphi)|$.

In continental areas or over land, apart from some regions (e.g. Death Sea area), the geoid is located inside the masses, whereas the gravity potential $W$ is only harmonic outside the mases. Therefore, in order to calculate the geoid undulation from the potential $W$, a correction due to the masses over the geoid has to be applied which could be done by using a representation of the topography in terms of spherical harmonics, $\bar{C}_{lm}^{topo}$ and $\bar{S}_{lm}^{topo}$. Using the model spherical harmonic coefficients from potential and topography, the geoid can be expressed as first approximation by:

$$
\begin{aligned}
N(\varphi,\lambda) \approx & \frac{GM}{r_e \gamma(r_e,\varphi)} \sum_{l=0}^{l_{max}} \left(\frac{R}{r_e}\right)^l \sum_{m=0}^{l} \bar{P}_{lm}(\sin\varphi)\left(\bar{C}_{lm}^{'} \cos m\lambda + \bar{S}_{lm}^{'} \sin m\lambda\right) \\
& - \frac{2\pi G\rho}{\gamma(r_e,\varphi)}\left[R\sum_{l=0}^{l_{max}}\sum_{m=0}^{l} \bar{P}_{lm}(\sin\varphi)\left(\bar{C}_{lm}^{topo}\cos m\lambda + \bar{S}_{lm}^{topo}\sin m\lambda\right)\right]^2
\end{aligned}
\qquad \textbf{Eq. (11)}
$$

whereas the gravity disturbance can be approximated by its radial component:

$$
\delta g(r,\varphi,\lambda) \approx \frac{GM}{r^2}\left[\sum_{l=0}^{l_{max}}\left(\frac{R}{r}\right)^l (l+1)\sum_{m=0}^{l} \bar{P}_{lm}(\sin\varphi)\left(\bar{C}_{lm}^{'} \cos m\lambda + \bar{S}_{lm}^{'} \sin m\lambda\right)\right]. \quad \textbf{Eq. ( 12)}
$$

In the following section, we will give examples of different functionals and their relevant applications. For the details of the formulations and insight to the other functionals, the reader is referred to Barthelmes, 2013.

### 2.2.1 Static global gravity field models of the Earth

As mentioned above, the ICGEM Service was established 15 years ago to mainly collect all available static global gravity field models from different institutions under one umbrella and make these models freely available to the public. Therefore, this feature is the fundamental component of the service and special attention is paid to maintain the complete list of static global gravity field models with a possibility to assign DOI number upon model developer(s)'s request. The three main complementary data sources to compute gravity field models are satellite based and terrestrial gravity measurements and satellite radar altimetry. The satellite based gravity measurements cover long wavelength information of the gravity field, whereas the spatial details of the gravity field (i.e. short wavelengths or high frequencies) are collected via terrestrial, airborne and shipborne gravity measurements and radar altimetry. The altimetry records yield the sea surface height which after some corrections can be taken as mapping of the geoid over the oceans and seas (e.g. Rummel and Sanso, 1993). Consequently, high degree and order resolution static gravity field models can only be developed based on a combination of the three sources which can also be supported by high resolution topographical models. These high-resolution static gravity field models are





used for regional geoid and gravity field determination in geodesy and geophysics, as well as for geodynamic interpretation and modelling.

The development of the Earth Gravitational Model 2008 (EGM2008) (Pavlis et al., 2012) was a very important milestone in terms of delivering high resolution static global gravity field models. The spherical harmonic degree and order expansion

reached up to 2159 using a combination of available gravity and topography data available worldwide. The improvement was due to the introduction of the National Geospatial-Intelligence Agency (NGA)'s worldwide terrestrial data coverage. After the release of EGM2008, different processing centres were also able to take the advantage of using the EGM2008 grids for the higher frequency components of the gravity field and to develop different "combined" high resolution global gravity field models. One of the high resolution static global gravity field models developed by GFZ is EIGEN-6C4 (Förste et al., 2014,

Förste et al., 2016b) which is also a combination of satellite and terrestrial data and expanded up to spherical harmonic degree and order 2190. EIGEN-6C4 contains data from the GOCE mission which was not yet available for EGM2008. With the latest release of EIGEN series, the spatial resolution of the "Potsdam Gravity Potato" has been resolved up to ~9 km half wavelength.

The global geoid undulations and gravity disturbances computed from EIGEN-6C4 are shown in Figures 7 and 8, respectively. Even though the Earth's interior is still a mystery, gravity can help to understand what is inside our planet. Regions inside the

15 Earth with higher mass densities (with respect to the mean density) produce larger gravity attraction on the surface, and mass deficit cause lower gravity. However, the influence of density anomalies on the geoid (which describes the gravity potential) is, at first glance, not so obvious. Positive density anomalies result in positive geoid undulations "geoid bumps" (i.e. the mean sea surface which follows the geoid is positive with respect to the ellipsoid) even though the gravity attraction is stronger. If we bear in mind that all the gravity forces must be perpendicular to the geoidal surface, this becomes apparent. Also, if we

"switch on" a (spherical) positive density anomaly beneath the ocean, the water flows towards the stronger attraction till the forces are perpendicular to the surface (i.e. no tangential forces are left) which is the case in the region of the North Atlantic. On the contrary, the big geoidal "dale" (negative geoid undulation) in the Indian Ocean is the result of mass deficit underneath the Earth (Ghosh et al., 2017). In that sense, the use of gravity field models to supplement the geophysical and geological models enhance our understanding of the Earth's dynamics.

The studies on the development of high resolution global gravity field continue with the reprocessed GOCE and GRACE data. Moreover, National Geodetic Survey (NGS) has collected plenty of new terrestrial data in the US (e.g. GRAV-D project) (Li et al., 2016) and worldwide; and it is expected that the new EGM model from NGA will be available in 2020. An experimental model as the precursor study for the upcoming EGM2020, namely XGM2016 has already been released in 2016 with the degree and order expansion of 719 (Pail et al., 2018). It is expected that the EGM2020 will have a spatial resolution of about

9km or better.



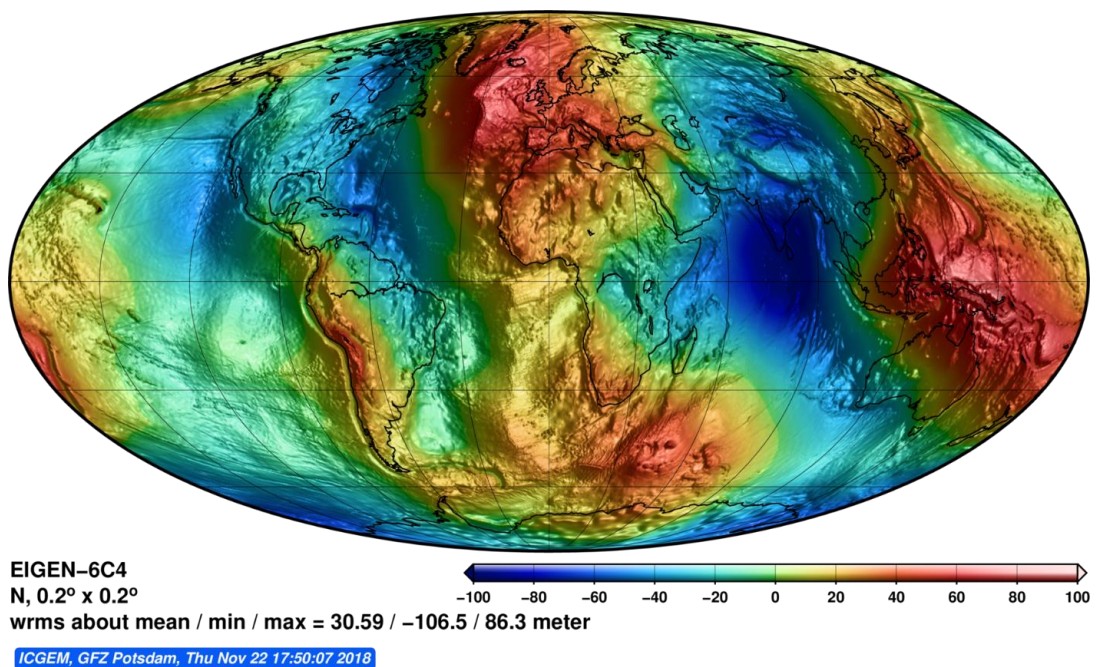

**Figure 7: Geoid undulation computed from EIGEN-6C4 combined gravity field model expanded up to degree and order 2190.**

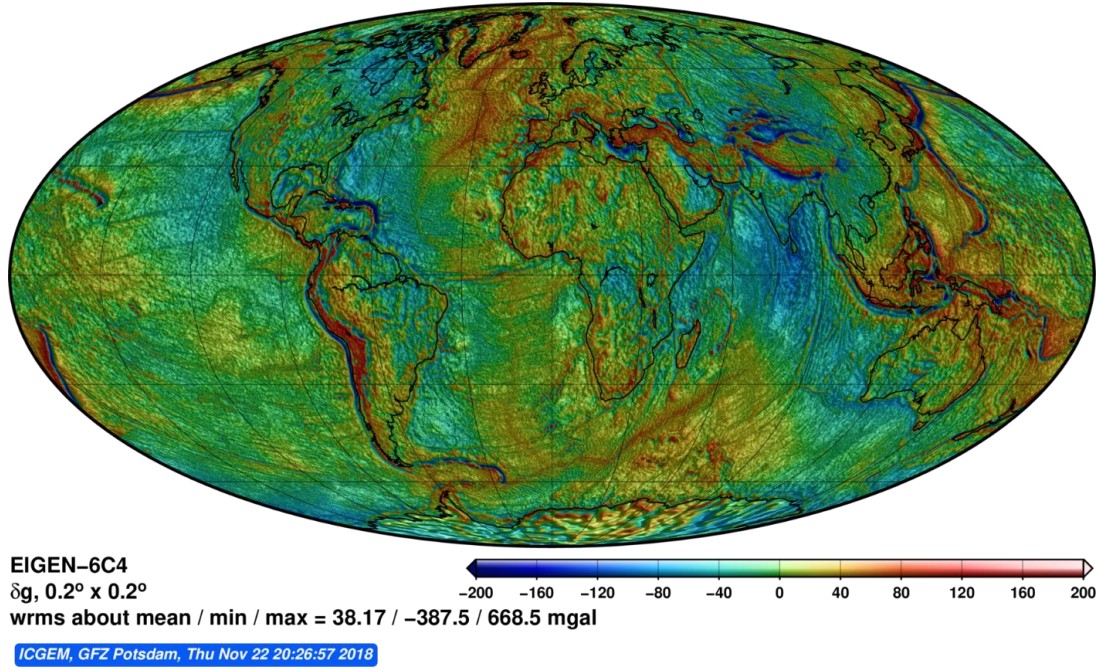

5    **Figure 8: Gravity disturbance computed on the Earth surface from EIGEN-6C4 combined gravity field model expanded up to degree and order 2190.**



A lower degree expansion of a gravity field model simply means a lower resolution in the spatial domain. The refined features of the gravity field are only visible using the high degree and order coefficients. Figure 9 shows four examples of different degree expansions, 50, 150, 250 and 500 of EIGEN-6C4 gravity anomalies corresponding to about 400, 133, 80 and 40 km half wavelength spatial resolution, respectively. It becomes obvious that the features are refined in spatial domain more and

5 more not only in the land but also over the oceans as the model is expanded up to higher degree and order. Accordingly, the ultimate goal would be the development of high resolution and high quality static gravity field model taking advantage of different datasets available.

As mentioned in the introductory section, gravity field models are important model inputs in several research fields. In geodesy, they are most commonly used for the GNSS levelling. Together with a high quality and high resolution geoid model, ellipsoidal

10 heights (geometric heights) measured using GNSS sensors can provide physical heights very efficiently. In the past, the physical heights were measured via spirit levelling (or other levelling methods) which has been limited to the road ways and widely accessible areas only. Other application areas of static gravity field models together with temporal and topographic gravity field models are summarized in Table 1. The complete list of the currently available 168 static gravity field models together with their expansion degrees are given in Appendix 2.





**Figure 9:** Geographical distribution of gravity anomalies in mGal with different spectral ($l_{max}$) and spatial resolution (half wavelength $\lambda_{min}/2$) of EIGEN-6C4. Spherical harmonic degree expansions for the four examples are as follows: a) 50 b) 150 c) 250 d) 500 which correspond to 400, ~133, 80 and 40 km half wavelength spatial resolution, respectively. See the refined features and better spatial localisation as the model expansion increases. For instance, topographical features such as mountains are well resolved in Figure 9d, whereas they are not precisely located in Figure 9a. The transmission borders of higher and lower anomalies in Alps and Mediterranean Sea are better resolved in Figure 9d, whereas it is not possible to distinguish and locate them precisely in Figure 9a. Note the different colour scales which have not been changed and kept as retrieved from ICGEM.



## 2.2.2 Temporal global gravity field models

Using the models derived from input data of dedicated time periods, it is possible to monitor the temporal changes in the gravity field. The spatial coverage of the shorter period observations is not as dense as in the longer periods. Therefore, the spatial resolution of temporal gravity field models (~ 160 km) is lower than those of the static gravity field models (~ 9km).

However, on the contrary to the static gravity field models, a mean over a short time provides a higher resolution in the time domain (e.g. 10 days, 1 month).

Both, the GRACE and now GRACE-FO missions are fundamental in observing the variation of the global gravity field. There are three official data analysing centres for GRACE and GRACE-FO data, namely GFZ, JPL (Jet Propulsion Laboratory) and UTCSR (University of Texas Center for Space Research), which calculate temporal global gravity field models within the

missions Science Data System. Even though the software packages of the three analysis centres are independent; they use the same Level 1 data (raw measurements from the satellite that are converted into engineering units, Level 1A) and edited and down-converted data (Level-1B) as input, nearly identical processing standards and background models to generate the GRACE/GRACE-FO Level 2 products (e.g. spherical harmonic coefficients for monthly periods). Same processing standards (Bettadpur 2012, Dahle, 2012; and Watkins and Yuan, 2014) mean common properties of the data processing (e.g. removing

solid Earth tides or non-tidal atmospheric and oceanic effects from measurements). After the well-known effects of other geophysical phenomena (e.g. air pressure, tides) are removed, the residuals are mainly expected to represent the water mass redistributions over a certain time period. However, mathematical methods including instrument parameterisations applied in designing measurement equations or Level-1B data editing and weighting vary among the three centres and this results in slightly different model coefficients. For the visualisation of the GFZ Level-2 solutions and access to use-ready gridded Level-

3 data, the reader is referred to the Gravity Information Service (GravIS) platform (http://gravis.gfz-potsdam.de/home).

The three data analysing centres release unconstrained solutions which means that no data besides GRACE measurements are applied nor any regularization (sometimes called stabilization) is used in the solution. After the solutions are retrieved, the lower degree ($C_{20}$) component of the temporal gravity field from GRACE/GRACE-FO is replaced with higher accuracy values derived from the SLR measurements. The disadvantage of these unconstrained models is the fact that the high-degree

coefficients have large errors (e.g. from aliasing of tidal and non-tidal mass variations or errors in the satellite-to-satellite tracking) and they are not recommended to be used directly (i.e. without filtering). On the other hand, users are free to develop their own filters or apply the commonly used DDK filters (Kusche et al., 2009), which are also offered in the ICGEM Calculation Service.

Temporal gravity field models developed by different institutions and agencies are summarized in Appendix 3. Even though

the initial models are derived based on the monthly coverage of GRACE observations; recently, daily models computed using the state of art techniques are published via the ICGEM Service (Mayer-Gürr et al., 2018). Moreover, combinations of different



measurements from different satellites (SLR normal matrices from 9 satellites and position data for CHAMP and GRACE) are used to derive monthly solutions (Weigelt et al., 2013) and also included in the ICGEM monthly series database as well.

Each temporal gravity model has different characteristics and may help retrieving different information depending on its data content and the application area it is used for. For instance, monthly models are very useful and important in monitoring the variations in the terrestrial hydrological cycle (Schmidt et al., 2006), ice melting (Velicogna, 2009), sea level change (Cazenave et al., 2009) and to help investigating climate change related variations in the Earth system (Wahr et al., 2004), whereas daily solutions have the potential to be used to monitor short term scale variations such as flood events and they contribute assessing natural hazards as proven with the successful outcomes of the EGSIEM project (Gouweleeuw et al., 2018). The results are generally presented in terms of equivalent water height or water column (Wahr et al., 1998, Wahr 2007). Some examples on the temporal gravity field models are shown in Section 3.

### 2.2.3 Topographic global gravity field models

Topographic global gravity field models are one of the most recent products that are included in the ICGEM Service. They represent the gravitational potential generated by the attraction of the Earth's topographic masses and enrich the possible applications of the geopotential models in geodesy and geophysics (Hirt and Rexer, 2015; Grombein et al., 2016; Hirt et al., 2016; Rexer et al., 2016). Different than the satellite based or combined gravity field models, gravity from these models is computed based on very high resolution digital elevation models which describe the shape of the Earth and model of mass densities inside the topography; therefore, they are not based on real gravity measurements.

This type of models are also called synthetic gravity models or forward models. Topographic masses used in the forward modelling include not only all solid Earth topography (rock, sand, basalts, etc.) but also ocean and lake water and ice sheets. These models can help retrieving very high frequency components of the global gravity field, interpret and validate real gravity measurements and global gravity field models, and help filling the gaps in which the actual gravity measurements are limited or not available, as it is the case in EGM2008 (Pavlis et al., 2012). More importantly, they can be used to subtract the topographical gravity signal from the gravity measurements and model computed gravity data; and make any other gravity signal visible that are related to the inner Earth. Therefore, use of these models is becoming more important in all kind of geophysical applications. An example of topographic model computed gravity anomalies in the Antarctica region together with the EGM2008 disturbances in the same area are shown in Figure 10. Note the resolved features in topographic model dV_ELL_Earth2014_plusGRS80 due to the availability of the high resolution topography data in the area. The typical applications for using these models are given in Table 1 and a list of currently available topographic gravity field models on the ICGEM website is provided in Appendix 4.



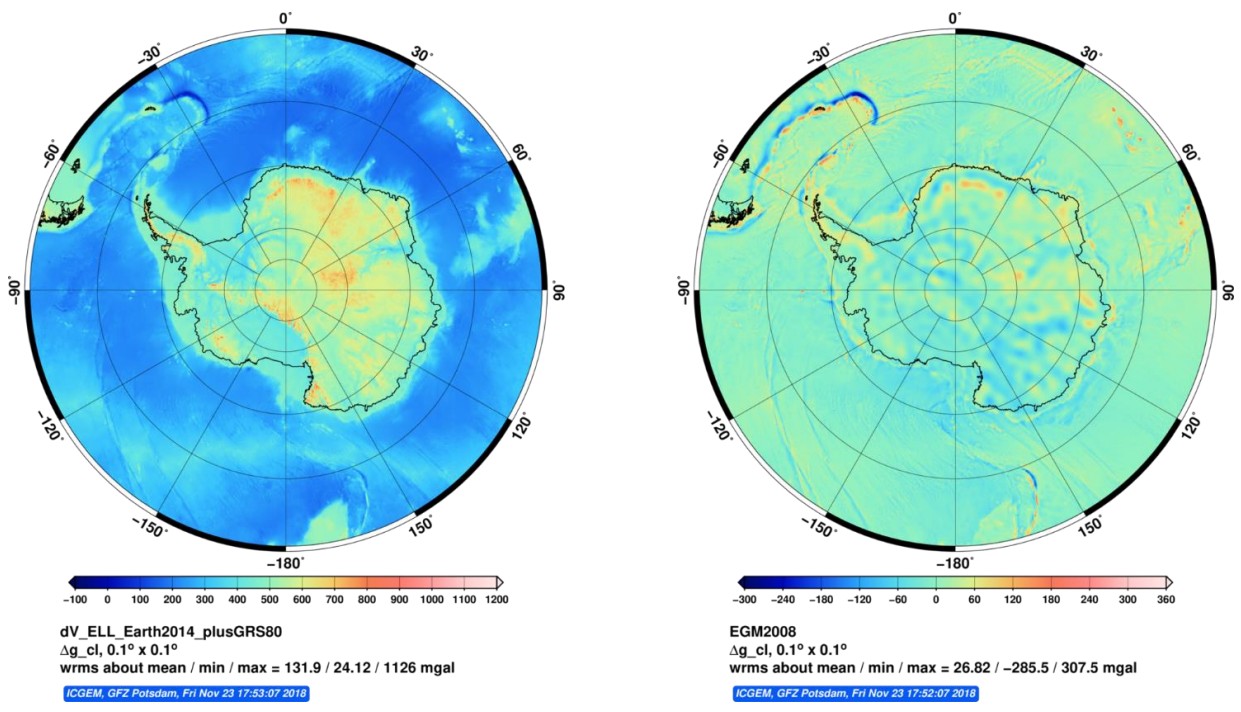

**Figure 10: The classical gravity anomalies which are also known as free air gravity anomalies computed on the Earth's surface based on a) topographic model dV_ELL_Earth2014_plusGRS80 and b) EGM2008. It is clearly seen that the features in Antarctica are better resolved in a. Note that the scale is left as is for individual cases on purpose since the current shape of the figures are what exactly the ICGEM calculation service provides.**

**Table 1: The application areas of the Global Gravity Field Models.**

| Application of gravity field models | | |
|---|---|---|
| **Static Gravity Field Models** | **Temporal Gravity Field Models** | **Topographic Gravity Field Models** |
| Geodesy<br>• Regional geoid modelling (using remove-compute-restore technique)<br>• Definition of a unique vertical datum and height modernization<br>• Satellite orbit determination | Geodesy<br>• Monitoring the change in the static gravity field model<br>• Monitoring the change in the regional geoid model | • Modelling high-resolution gravity field assuming the highest spatial resolution features are mainly produced by topography<br>• Modelling the omission error of the gravity field models<br>• Evaluation of satellite-based gravity field models using external independent data<br>• Reducing terrain and topographic gravity to smooth gravity measurements<br>• Reducing topographic gravity to retrieve gravity signals of other sources<br>• Modelling Bouguer gravity anomaly |
| Oceanography<br>• Monitoring sea level variation<br>• As a reference surface for sea surface topography<br>• Derivation of geostrophic ocean surface currents | Hydrology<br>• Monitoring inter-annual, seasonal and sub seasonal water level variations<br>• Monitoring ground water variations<br>• Monitoring ice melting | |
| Geophysics<br>• Monitoring mass and density distribution, isostasy, mantle processes | Atmosphere<br>• Monitoring inter-annual, seasonal variations | |
| | Geophysics<br>• Monitoring post glacial rebound<br>• Detection of co- and post- seismic mass redistribution | |



### 2.2.4 Models of other celestial bodies

Gravity field models for other celestial bodies, the Moon, Venus and Mars are by-products of the ICGEM Service. They are provided due to the interest of the model developers and users. These models have the same mathematical representation, i.e.

expansion of spherical harmonic series, as the static gravity field models of the Earth. Therefore, it is easy to include also these models in the ICGEM calculation and visualisation services.

These models are also developed based on similar observations of the gravity field of the body. For instance, spacecraft to spacecraft tracking observations from the Gravity Recovery and Interior Laboratory (GRAIL) have been used to develop a gravitational field of the Moon (Zuber et al., 2013).

**3 Services of ICGEM**

### 3.1 Calculation Service

By the time that the ICGEM Service was established, it was naturally installed together with the calculation and visualisation services. Due to the interest of scientists and students worldwide, the ICGEM team have developed a web interface to calculate gravity field functionals (e.g. geoid undulation, height anomaly, gravity anomaly) from the spherical harmonic representations

of the Earth's global gravity field on freely selectable grids with respect to a reference system of user's preference. This service is the only online service worldwide available that computes variety of gravity field functionals with the GMT plots (Wessel et al., 2013) provided for grid values and the option to download the computed values. During the 15 years, interested researchers and students have used the ICGEM service extensively for calculating gridded gravity field functionals (see Figure 11).

Starting from December 2018, the ICGEM Service introduced the calculation of gravity field functionals also at the user defined list of points. The list of the particular points can be prepared by the user in one of the allowed formats and the calculations are performed directly at those particular points. Different heights for different points can be introduced in the point calculation which is different to the grid calculation where the height is assumed same for all the grid points and consequently delivers results faster. The point calculation service was developed based on the request from the users as well.

The list of the functionals has also been expanded during the years based on requests and their descriptions are given in Table 2. The equations referred in Table 2 are given in Barthelmes, 2013 in detail. Calculated results are not only provided in ASCII format but also visualised using the Generic Mapping Tools (GMT) software (Wessel and Smith, 1998; Wessel et al., 2013) with the basic statistics provided. An example can be found in Figure 12.

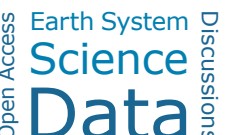

For the point calculations, after the user uploads the text file of the set of data points in a predefined format, the points are displayed on the map. The example in Figure 13 shows the GNSS/levelling benchmark points in Europe which also are used in the geoid comparisons in the model evaluations.

Some of the gravity field functionals are calculated based on a particular reference system in 3D while some others depend on 2D position only. For the 3D functionals, the ICGEM calculation service provides different options for the reference system such as the commonly used ones as WGS84 (World Geodetic System 1984) and GRS80 (Geodetic Reference System 1980). In addition, it provides users the option to define their own reference system by providing the radius, $GM$, flattening ratio, and angular velocity of the rotation ($\omega$). Considering the researchers working on different regions of the world based on different normal ellipsoid reference systems, this feature is very helpful and eliminates the time the user needs for the transformation between the reference systems.

Another component in the calculation of the gravity field functionals is the systematic effect that is due to the reference tide system regarding the flattening of the Earth. These are important for the definition of the geoid. Three different tide systems, namely tide-free, zero-tide and mean-tide systems (Lemoine et al., 1998), can be selected via the given options on the calculation page. It is worth reminding that in the:

- *Tide-free system,* the direct and indirect effects of the Sun and the Moon are removed.
- *Zero-tide system,* the permanent direct effects of the Sun and Moon are removed but the indirect effect related to the elastic deformation of the Earth is retained.
- *Mean-tide system,* no permanent tide effect is removed.

Geoidal surface is generally given in terms of geoid undulations or geoid heights with respect to a reference system that consists of a best approximating (in the least squares sense) geometric rotational ellipsoid (normal ellipsoid) and an associated best approximating ellipsoidal normal potential $U = U(r, \varphi, \lambda)$. The defining parameters of such a reference system also define the value of $U = U_0$ (ellipsoidal equipotential surface of the normal potential = geometrical ellipsoid of the reference system which approximates the Earth over the oceans). Moreover, the normal potential is defined in such a way that its value on the normal ellipsoid is $U = \text{constant} = U_0$ and approximates the real value $W_0$ as good as it is known (at the time when this reference system is defined). Hence, the reference system also defines the value $W_0 = U_0$. It is worth noting that an improvement of the numerical estimation of $W_0$ value is still under discussion and requires up to date information of small changes of gravity field potential (e.g. due to sea level rise).

Following up the above discussion, zero degree term arises when $W_0$ is chosen/calculated different than $U_0$ and/or when $GM$ values between the geopotential model and reference ellipsoid are different. Therefore, this term needs to be taken into





account to calculate the geoid undulation correctly with respect to a known reference ellipsoidal surface (see also question 16 in http://icgem.gfz-potsdam.de/icgem_faq.pdf).

**Table 2: The list of gravity field functionals available on the calculation service for grid and point calculations. The detailed description and definitions of the gravity field functionals are given in detail in Barthelmes (2013). The equation numbers (eqs.) refer to the same document.**

| Gravity field functional | Definition See also Barthelmes, F., (2013). | Static model | Temporal model | Topography related model | Celestial Object |
|---|---|---|---|---|---|
| height_anomaly | It is an approximation to geoid height according to Molodensky's theory, defined on the Earth's surface (eqs. 81 and 119) where the height (elevation) used in the calculation is taken from etopo1m automatically. | x | x | x | - |
| height_anomaly_ell | It is the generalized pseudo height anomaly which is defined on the ellipsoid; therefore, the h value used in the calculation is set to zero. | x | x | x | x |
| geoid | It is a particular equipotential surface of the gravity potential of the Earth, which is equal to the undisturbed sea surface and its continuation below the continents. Here it is approximated by the height anomaly plus a topography dependent correction term (eqs. 71, 117). | x | x | x | - |
| gravity_disturbance | The gravity disturbance is defined as the magnitude of the gradient of the potential at a given point on the Earth's surface minus the magnitude of the gradient of the normal potential at the same point. (eqs. 87, 121 − 124). | x | x | x | - |
| gravity_disturbance_sa | It is calculated by spherical approximation on h=0 or above at an arbitrary height over the ellipsoid, h>0. | x | x | x | x |
| gravity_anomaly | The gravity anomaly (according to Molodensky's theory) is defined as the magnitude of the gradient of the potential on the Earth's surface minus the magnitude of the gradient of the normal potential on the Telluroid (eqs. 101 and 121 − 124). | x | x | x | - |
| gravity_anomaly_cl | The classical gravity anomaly is defined as the magnitude of the gradient of the downward continued potential on the geoid minus the magnitude of the gradient of the normal potential on the ellipsoid (eqs. 93, 121 − 124). This type of gravity anomalies are also known as free-air gravity anomaly. | x | x | x | - |
| gravity_anomaly_sa | The gravity anomaly calculated by spherical approximation (eqs. 100 or 104, 126). Unlike the classical gravity anomaly, the Molodensky gravity anomaly and the spherical approximation can be generalised to 3D space, hence here it can be calculated on h=0 or above the ellipsoid, h>0. | x | x | - | - |
| gravity_anomaly_bg | The (simple) Bouguer gravity anomaly is defined by the classical gravity anomaly minus the attraction of the Bouguer plate. Here it is calculated by the spherical approximation of the classical gravity anomaly minus $2\pi G\rho H$ (eqs. 107, 126). The topographic heights H(λ,φ) are calculated from the spherical harmonic model of etopo1m up to the same maximum degree as the gravity field model. For H ≥ 0 (rock) → ρ = 2670 kg/m3, and for H<0(water) →ρ=(2670−1025) kg/m3 is used. The density contrast between ice and rock is not been taken into account. | x | x | - | - |
| gravity_earth | The gravity is defined as the magnitude of the gradient of the potential (including the centrifugal potential) at a given point. Here it will be calculated on the Earth's surface (eqs. 7, 121 − 124). | x | x | x | - |
| gravity_ell | The magnitude of the gradient of the potential calculated on/above the ellipsoid including the centrifugal potential (eqs. 7, 121−124). | x | x | x | - |
| potential_ell | The potential of the gravity field of the Earth without the centrifugal potential (gravitational field). Here it can be calculated on/above the ellipsoid (eq. 108). | x | x | x | x |
| gravitation_ell | The magnitude of the gradient of the potential calculated on or above the ellipsoid without the centrifugal potential (eqs. 7, 122). | x | x | x | x |



| second_r_derivative | The second derivative of the disturbance potential in radial direction calculated on/above the ellipsoid. | x | x | x | x |
|---|---|---|---|---|---|
| water_column | The variable thickness of a fictitious water layer which is distributed over the reference ellipsoid and produce the disturbance potential or the geoid undulations. For calculating "water_column" from a gravity field model the elastic deformation of the Earth due to the load of the water layer is considered. | x | x | - | - |
| vertical_deflection_abs | This is the magnitude of the deflection of the vertical. It is the angle between the vector of gravity and the vector of normal gravity both at the same point (h,λ,φ). | x | x | - | - |
| vertical_deflection_ew | This is the east-west component of the deflection of the vertical. It is the east-west component of the angle between the vector of gravity and the vector of normal gravity both at the same point (h,λ,φ). | x | x | - | - |
| vertical_deflection_ns | This is the north-south component of the deflection of the vertical. It is the north-south component of the angle between the vector of gravity and the vector of normal gravity both at the same point (h,λ,φ). | x | x | - | - |



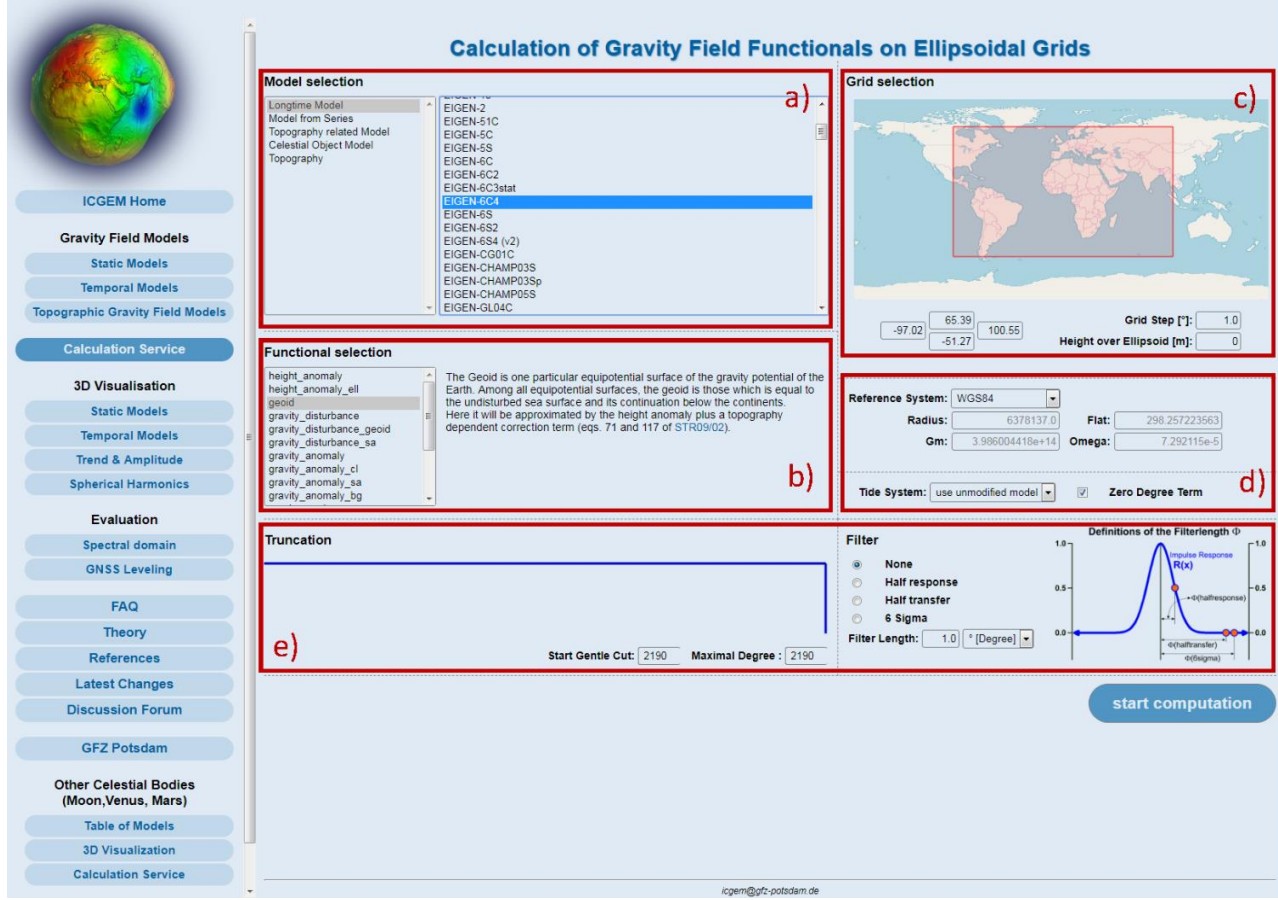

**Figure 11: Snapshot of the calculation service interface. The calculation settings allow the user to choose the model of preference from the list of global gravity field models (a), the functional of interest (b), the boundaries of the area and the grid interval (c), reference and tide system (d), and truncation degree and filtering (e) before starting the computation. The grid area can also be selected using the red rectangle in (c) by simply changing its borders or entering the coordinates manually. Moreover, grid interval can be entered in terms of degrees.**



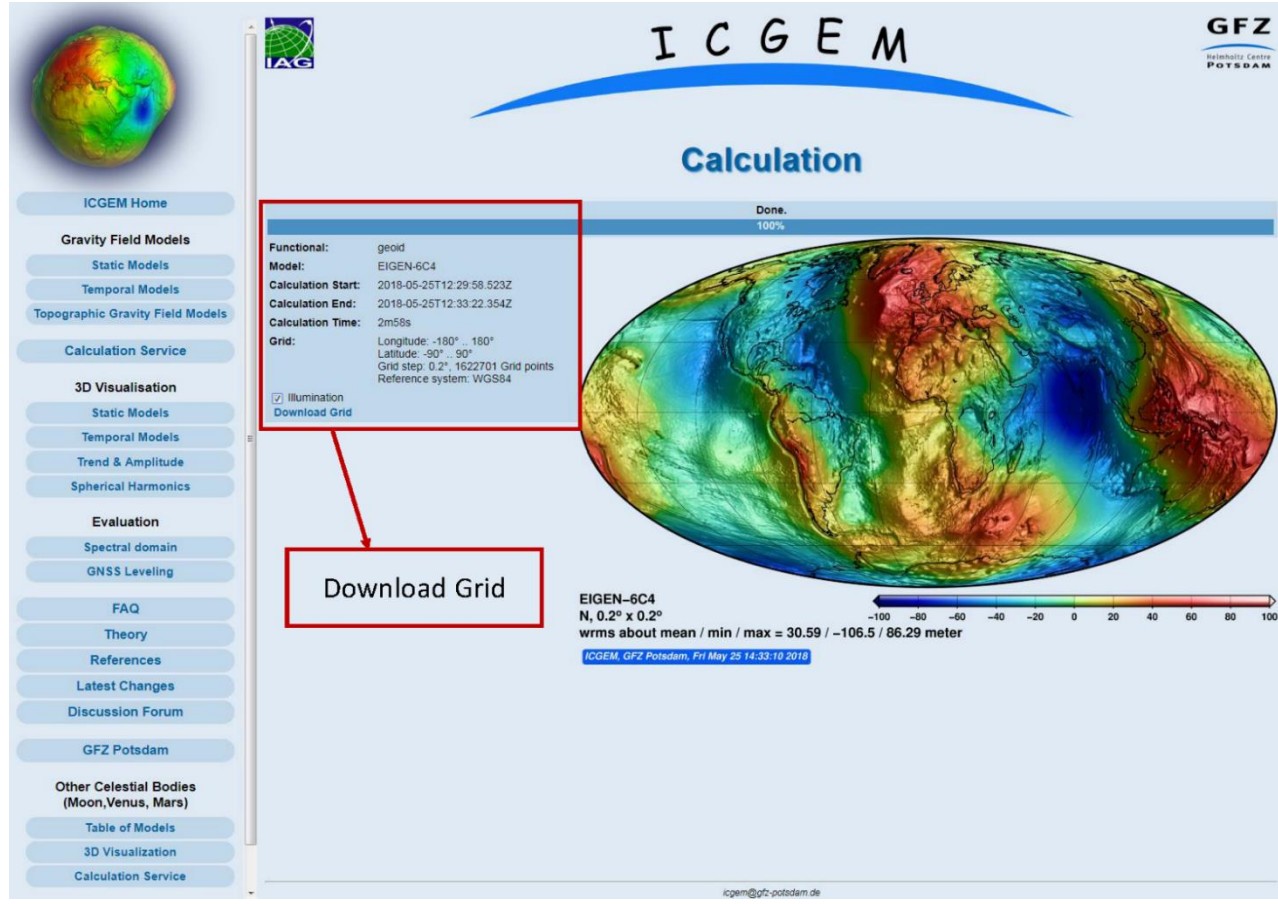

**Figure 12: Snapshot of calculation service interface with the results provided in numerical and map view. The figure and grid values can be downloaded from the same page.**





**Figure 13: Snapshot of the point value calculation service on ICGEM. The example shows the GNSS/levelling benchmark points in Europe that are used also in the ICGEM static model evaluation. The user can choose one, multiple or all the functionals at the same time to compute at those points.**



## 3.2 3D Visualisation Service

An online interactive visualisation service of the static gravity field models (geoid undulations and gravity anomalies), temporal models (geoid undulation and equivalent water column), trend and annual amplitude of GRACE gravity time variations, and spherical harmonics as illuminated projection on a freely rotatable sphere are available on the ICGEM Service.

The visualisation service was established to provide the users and researchers a sophisticated visual representation of the gravity field related products and it was the first of its kind at the time it was available to the general public with its well-known name "Potsdam Gravity Potato". It has become a service which is very useful for quick look analyses of the functionals globally and also for tutorial purposes of different educational levels (http://icgem.gfz-potsdam.de/vis3d/longtime). Users of this service can select the functional, the model, the grid interval and the spherical harmonic degree expansion of the model to

see the results on the 3D visualisation. Moreover, rotation tool can be used to locate different regions of the globe and the selected image can be exported via the export tool. An example of geoid undulation and gravity anomalies is shown in Figure 14.

Static model visualization enables also the demonstration of the differences between two models with a selected grid interval and spherical harmonic degree expansion. Zooming in and out functions are available which makes this tool very useful for

also advanced users to quickly investigate particular regions of interest based on different models quickly. Using the 3D visualisation service, as an example, the substantial differences between the new experimental geopotential model XGM2016 of the upcoming Earth Gravitational Model 2020 (EGM2020) and the older EGM2008 model are displayed for Antarctica and Himalaya regions (Figure 15). As shown in Figure 15a, the differences are mostly due to the 'terrestrial' update in Antarctica which are due to the updates of the 'non-data' or 'synthetic' values used in the EGM2008. Similar differences are also shown

for the Himalaya regions in Figure 15b.

3D Visualisation of temporal gravity field models displays computation of geoid undulation and equivalent water column (or equivalent water height) from different daily and monthly series with an option of using unfiltered or filtered model coefficients. The visualisation tool can also be used for animation purposes for different monthly series. Two different monthly series, January 2009 and May 2009, filtered using DDK1 filter are displayed in Figure 16a and 16b. The differences between

the two figures represent the mass redistributions. The visualisation of the trend and annual amplitude of GRACE measurements that are collected between 2002 and 2015 are also available as shown in Figure 16c and Figure 16d, respectively. The ice melting trend over Greenland, Glacial Isostatic Adjustment (GIA) effect in Alaska and Hudson Bay area, and the annual mass variation in the Amazon region which have been some of the priority research topics during the last few years can be clearly seen in these representations.




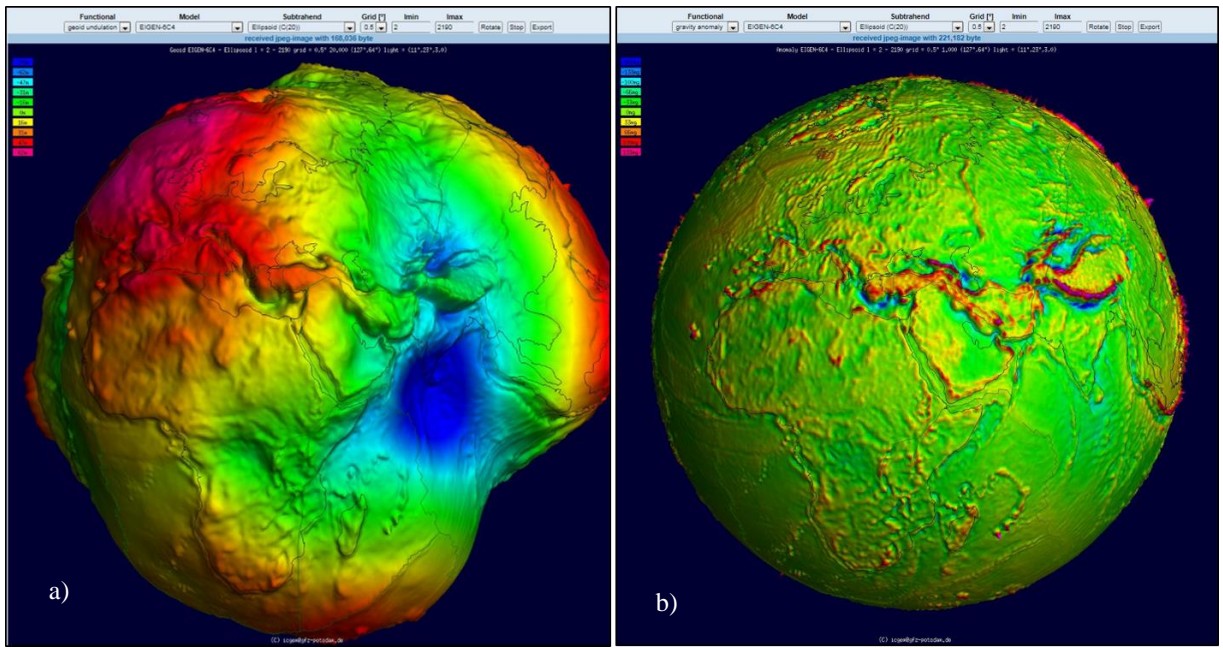

**Figure 14: Examples of visualisation service for a) geoid undulation and b) gravity anomaly computed from a high resolution combined static gravity field model.**

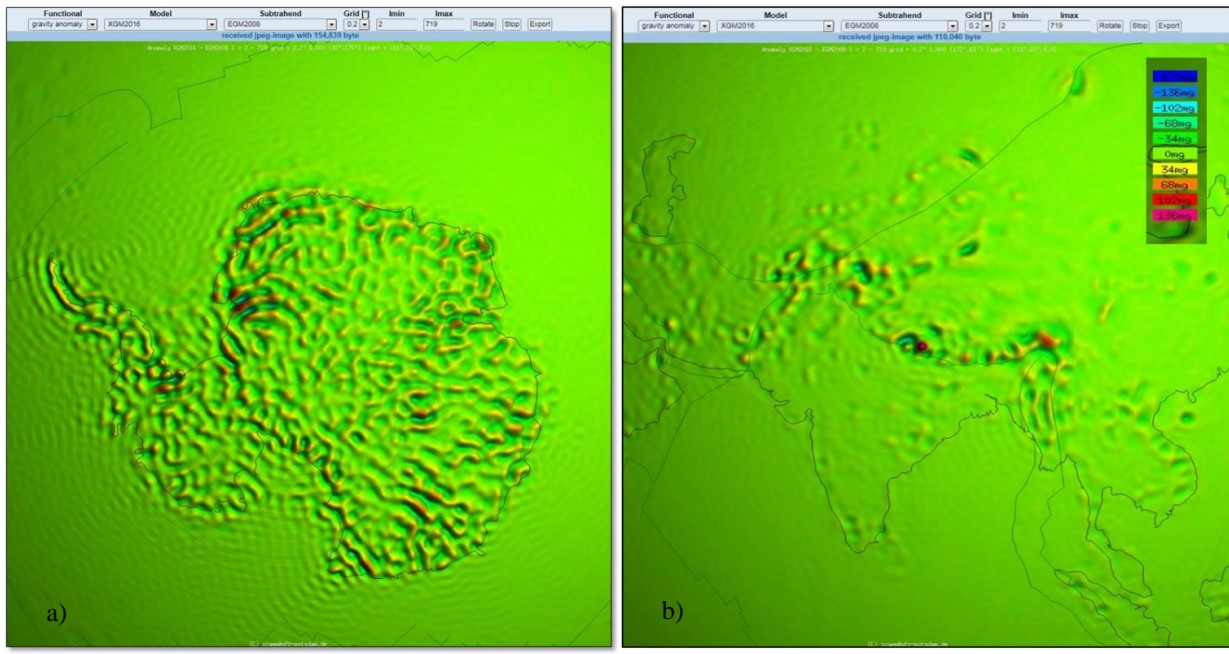

**Figure 15: Examples of visualisation service for the gravity anomaly differences computed from XGM2016 and EGM2018 up to degree and order 719 for the a) Antarctica and b) Himalaya regions. The EGM2008 relied exclusively on ITG-Grace03s (Mayer-Gürr, 2007) expanded up to d/o 120 to fill Antarctica with synthetic values; whereas, in XGM2016 (Pail et al., 2018), these synthetic values were derived from GOCO05s (Mayer-Gürr et al., 2015) and from forward modelling of ice and rock thicknesses from the Earth2014 Digital Terrain Model (Hirt and Rexer, 2015).**



**Figure 16: Snapshot of visualisation service for temporal models a) EWH in January 2009 b) EWH in May 2009, note that the EWH difference between the two months represents the mass distributions, c) trend, note the strong effect due to the GIA in Canada and Alaska and ice melting in Greenland d) annual amplitude, where the large signal amplitude in the Amazon region is noticeable.**



### 3.3 Evaluation of global gravity field models

With its additional evaluation service, ICGEM goes beyond the collection and distribution of the gravity field models. Before being published as part of the ICGEM Service, each new global model is investigated to ensure that its content is worthy to be published in the service. There are two techniques covered in the ICGEM evaluation procedure: 1) comparisons w.r.t. other

(already identified as reliable) global models in the spectral domain using signal degree amplitudes 2) comparison of the model calculated geoid undulations w.r.t. set of GNSS/levelling derived geoid undulations for different regions of the Earth.

### 3.3.1 Model evaluation with respect to other models in the spectral domain

One of the most commonly used techniques in the assessment of global gravity field models is the cumulatively looking at the signal and noise amplitudes per degree and signal and noise amplitudes. The signal can be computed using the spherical

harmonic coefficients whereas the noise can be computed using the associated errors. In the ICGEM evaluation procedure, we use the signal degree amplitudes, $\sigma_l$ of functional of the disturbing potential $T(r, \varphi, \lambda)$ at the Earth's surface but not the error degree amplitude, since not all of the models include the same type of error. Some of the models include formal errors whereas the other ones include calibrated errors. The signal degree amplitudes of the models can be computed by:

$$S_l = \sqrt{\sum_{m=0}^{l} \left( \overline{C}_{lm}^{'\,2} + \overline{S}_{lm}^{'\,2} \right)},$$  **Eq. (13)**

in terms of unit less coefficients. The outcomes refer to the internal accuracy of the global model in terms of geoid height, gravity anomaly or other functionals. The error degree variance can also be computed using the spherical harmonics associated error coefficients using the same formula (Eq. 13).

The outcomes of this analysis do not necessarily represent the model characteristics or signal to noise ratio of a particular area or a region but represent the model characteristics globally. In our comparisons,

we particularly use geoid heights signal amplitudes per degree which can be computed via:

$$\sigma_l(N) = R\sigma_l,$$  **Eq. (14)**

in terms of meter. An example of the comparison of two recent static global gravity field models, satellite only model GOCO05S and combined model EIGEN-6C4, is shown in Figure 17.





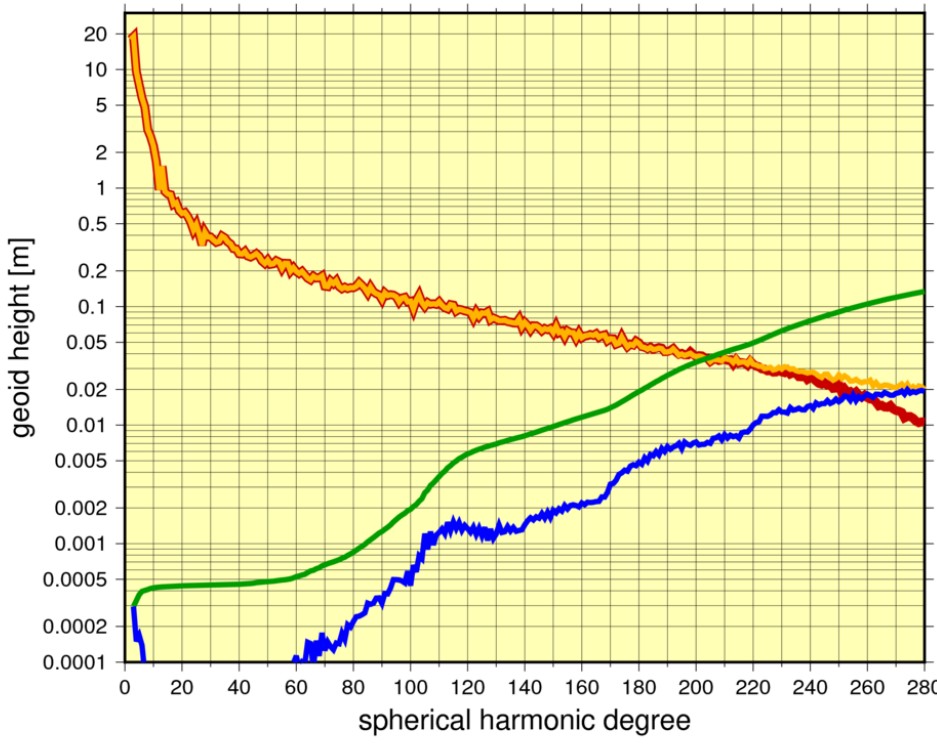

Figure 17: Spectral comparison of two static global gravity field models, GOCO05S and EIGEN-6C4. Note that the comparisons are performed for each degree separately. GOCO05S is a satellite only model, whereas the EIGEN6C4 is a combined model that uses both the satellite and terrestrial measurements. The blue curve represents the difference in the amplitude of the GOCO05S and EIGEN-6C4 combined static gravity field models per degree, whereas the green line represents the difference amplitudes as a function of maximum degree of the two models. See the increasing difference as the degree increases due to the contribution of the terrestrial gravity data to EIGEN-6C4.



### 3.3.2 Model evaluation with respect to GNSS/levelling derived geoid undulations

Another way of assessing a global gravity field model is to compare the model outputs with respect to external sources. For instance, it is very common to compare the model computed geoid undulations with GNSS/levelling-derived geoid undulations. Traditionally, geoid undulations have been derived from the ellipsoidal and orthometric heights that are measured using GNSS sensors and via levelling which is limited to the levelling benchmark points. This kind of evaluation is also valid for other gravity field functionals, such as gravity anomalies or deflections of vertical where the model computed values are compared with the terrestrial measurements. The advantage of this method is that it is suitable to assess the model outcomes in a regional level or at a particular area but the assessments are only as good as the quality of the external datasets used in the validation. ICGEM has collected some series of GNSS/levelling datasets from different countries. These countries are Australia, Brazil, Canada, Japan and the USA. Moreover, a series of data points from Europe is also included in the comparisons. More information on the GNSS/levelling data points provided in Table 3.

In contrast to the global gravity field models, these data are not freely available. Their availability is limited to the legal restrictions or the observers' own interest. Due to the relevance of these external data sets for the model evaluation, ICGEM will address this issue and develop strategies in improving the availability of these data for the general public and for the benefit of the global community.

In general, the GNSS/levelling measurements are collected over decades. Besides the epoch differences among the measurements, different GNSS or GPS equipment are used and different length of observations, and observing procedures are followed which cause the estimation accuracy of the ellipsoidal heights ($h$) to vary. The GNSS/levelling derived geoid undulations can be computed via

$$N = h - H,$$   **Eq. (15)**

which are erroneous. These errors are not taken into account in our assessments and obviously the comparison results will only be as good as the quality of both resources, global gravity field model and GNSS/levelling derived geoid undulations. Moreover, to perform realistic and informative comparisons, one needs to consider the omission error that is caused by the truncation of the global solutions. To ensure fair comparisons, the GNSS/levelling derived geoid undulations should also be reduced to the same spectral content of the gravimetric geoids, which is also not taken into account in our quick check assessments since the purpose of this service is to provide relative comparisons among the models. Comparison results are given in the evaluation section of the service and results for the recent models are shown in Figure 18.





**Table 3: Information on the GNSS/levelling benchmark points ICGEM collected during the years and corresponding authors/institutions.**

|  | Australia | Brazil | Canada | Europe | Japan | USA |
|---|---|---|---|---|---|---|
| # of points | 201 | 1112 | 2691 | 1047 | 816 | 6169 |
| Corresponding author | G. Johnston | D. Blitzkow, A. Cristina, O. Cancoro de Matos | M. Veronneau, February 2003 | Ihde et al., 2002 | Tokuro Kodama, | Milbert, 1998 |
| Corresponding Institution | Geoscience Australia | CENEGEO, the data belongs to the LTG/USP and the (IBGE) | NRCan |  | Geospatial Information Authority of Japan |  |

5   CENEGEO:  Centro de Estudos de Geodesia; IBGE: Brazilian Institute of Geography and Statistics; LTG/ USP: Laboratory of Topography and Geodesy/University of Sao Paulo; NRCan: Natural Resources Canada

| Nr | Model | Nmax | Australia (201 points) | Brazil (1112 points) | Canada (2691 points) | Europe (1047 points) | Japan (816 points) | USA (6169 points) | All (12036 points) |
|---|---|---|---|---|---|---|---|---|---|
| 168 | Tongji-Grace02k | 180 | 0.432 m | 0.592 m | 0.475 m | 0.587 m | 0.661 m | 0.525 m | 0.5357 m |
| 167 | SGG-UGM-1 | 2,159 | 0.217 m | 0.446 m | 0.13 m | 0.121 m | 0.076 m | 0.245 m | 0.2353 m |
| 166 | GOSG01S | 220 | 0.359 m | 0.518 m | 0.373 m | 0.426 m | 0.526 m | 0.442 m | 0.4392 m |
| 165 | IGGT_R1 | 240 | 0.317 m | 0.513 m | 0.348 m | 0.387 m | 0.483 m | 0.412 m | 0.4111 m |
| 164 | IfE_GOCE05s | 250 | 0.337 m | 0.512 m | 0.329 m | 0.385 m | 0.48 m | 0.414 m | 0.4081 m |
| 163 | GO_CONS_GCF_2_SPW_R5 | 330 | 0.33 m | 0.511 m | 0.299 m | 0.346 m | 0.442 m | 0.396 m | 0.3873 m |
| 162 | GAO2012 | 360 | 0.293 m | 0.531 m | 0.309 m | 0.453 m | 0.759 m | 0.366 m | 0.4177 m |
| 161 | XGM2016 | 719 | 0.218 m | 0.44 m | 0.151 m | 0.14 m | 0.125 m | 0.263 m | 0.2489 m |
| 160 | Tongji-Grace02s | 180 | 0.452 m | 0.605 m | 0.478 m | 0.596 m | 0.669 m | 0.53 m | 0.5417 m |
| 159 | NULP-02s | 250 | 0.351 m | 0.512 m | 0.375 m | 0.413 m | 0.508 m | 0.427 m | 0.4284 m |
| 158 | HUST-Grace2016s | 160 | 0.489 m | 0.658 m | 0.594 m | 0.69 m | 0.837 m | 0.596 m | 0.6273 m |
| 157 | ITU_GRACE16 | 180 | 1.778 m | 6.645 m | 1.591 m | 1.307 m | 0.976 m | 2.741 m | 2.9603 m |
| 157 | ITU_GRACE16 (upto130) | 130 | 0.515 m | 0.747 m | 0.676 m | 0.871 m | 1.093 m | 0.692 m | 0.7419 m |
| 156 | ITU_GGC16 | 280 | 0.335 m | 0.505 m | 0.31 m | 0.343 m | 0.45 m | 0.398 m | 0.39 m |
| 155 | EIGEN-6S4 (v2) | 300 | 0.327 m | 0.507 m | 0.298 m | 0.345 m | 0.447 m | 0.405 m | 0.3915 m |
| 154 | GOCO05c | 720 | 0.221 m | 0.445 m | 0.154 m | 0.138 m | 0.217 m | 0.262 m | 0.2541 m |
| 153 | GGM05C | 360 | 0.239 m | 0.461 m | 0.213 m | 0.225 m | 0.282 m | 0.321 m | 0.3055 m |
| 152 | GECO | 2,190 | 0.216 m | 0.451 m | 0.131 m | 0.123 m | 0.08 m | 0.246 m | 0.2371 m |
| 151 | GGM05G (upto210) | 210 | 0.357 m | 0.521 m | 0.374 m | 0.454 m | 0.543 m | 0.448 m | 0.4461 m |
| 151 | GGM05G | 240 | 0.326 m | 0.502 m | 0.342 m | 0.384 m | 0.487 m | 0.407 m | 0.4065 m |
| 150 | GOCO05s | 280 | 0.335 m | 0.505 m | 0.308 m | 0.344 m | 0.45 m | 0.399 m | 0.3904 m |
| 149 | GO_CONS_GCF_2_SPW_R4 | 280 | 0.322 m | 0.508 m | 0.33 m | 0.375 m | 0.473 m | 0.406 m | 0.4023 m |
| 148 | EIGEN-6C4 | 2,190 | 0.212 m | 0.446 m | 0.126 m | 0.121 m | 0.079 m | 0.247 m | 0.2361 m |
| 147 | ITSG-Grace2014s | 200 | 1.175 m | 1.273 m | 0.871 m | 0.962 m | 0.932 m | 1.095 m | 1.0468 m |
| 146 | ITSG-Grace2014k | 200 | 0.433 m | 0.611 m | 0.419 m | 0.582 m | 0.651 m | 0.542 m | 0.5347 m |
| 145 | GO_CONS_GCF_2_TIM_R5 | 280 | 0.336 m | 0.505 m | 0.31 m | 0.343 m | 0.45 m | 0.398 m | 0.39 m |
| 144 | GO_CONS_GCF_2_DIR_R5 | 300 | 0.327 m | 0.507 m | 0.299 m | 0.345 m | 0.447 m | 0.405 m | 0.3915 m |

The table is interactively re-sortable for all columns by clicking in the header cells.

10   **Figure 18: RMS of the mean differences between the model computed geoid undulations and the GNSS/levelling derived geoid undulations. The comparison results are shown for the most recent models and retrieved from http://icgem.gfz-potsdam.de/tom_gpslev.**

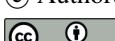



### 3.5 DOI Service

For more than a decade, the need for and value of open data have been expressed in major science society position statements, foundation initiatives, and in statements and directives from governments and funding agencies worldwide. The citable data publication with assigned digital object identifiers (DOI) can be regarded as best practice for addressing these requirements.

Ideally, the data are technically described and provided with standardised metadata, including the licence for reuse that are readable for humans and machines. Today, datasets with assigned DOI are fully citable research products that can and should be included in reference lists of research articles (Data Citation Synthesis Group 2014, Hanson et al., 2015).

Following the bottom-up structure of ICGEM, the DOI Service was developed as a request by the user community. Global gravity field models are often shared through ICGEM months or even years before they are described in scientific research

articles. The publication of static and temporal models with a DOI makes them citable and provides credit for the originators already with their publication via the ICGEM Service. The DOI Service of ICGEM was developed in cooperation with GFZ Data Services, the domain repository for the geosciences hosted by GFZ (http://dataservices.gfz-potsdam.de/portal/). To reduce the heterogeneity in the data documentation for static global gravity field models, we have developed standardised metadata templates for describing the models and cross-reference the model data with research articles, data reports with

detailed model description and other text or data publications. At the moment, all models with assigned DOIs are published under the Creative Commons Attribution 4.0 International Licence (CC BY 4.0).

For DOI-referenced models, data access is also provided via specific, ICGEM branded, DOI Landing Pages as shown in Figure 19, the GFZ Data Services catalogue or via the ICGEM Website (see Figure 20). In addition, on the DOI Landing Pages we provide direct links to the ICGEM Visualisation and Calculation Services for the specific model. The citation of the model

and the licence (we recommend to use Creative Commons Attribution 4.0 International CC BY 4.0) is included in the header of the data files themselves.

Since its implementation in late 2015, we have assigned DOIs to 17 static and 3 temporal series, mostly timely related to their first publication via ICGEM. As DOI-references datasets are required to remain unchanged, for the case of a model update we have developed a DOI versioning service with direct links between the two versions and a version history explaining the

differences (e.g. Förste et al., 2016a and b). GFZ Data Services provide their DOI landing pages and metadata for each dataset in machine-readable form (schema.org and XML, respectively), following DataCite 4.0 and ISO19115 metadata standards and is equipped with an Application Programing Interface (OAI-PMH) that allows automatic metadata exchange. Consequently, metadata from ICGEM models is also findable in the catalogues of e.g. DataCite (http://search.datacite.org/), B2Find (http://b2find.eudat.eu/) and the newly released Google Dataset Search engine (https://toolbox.google.com/datasetsearch).



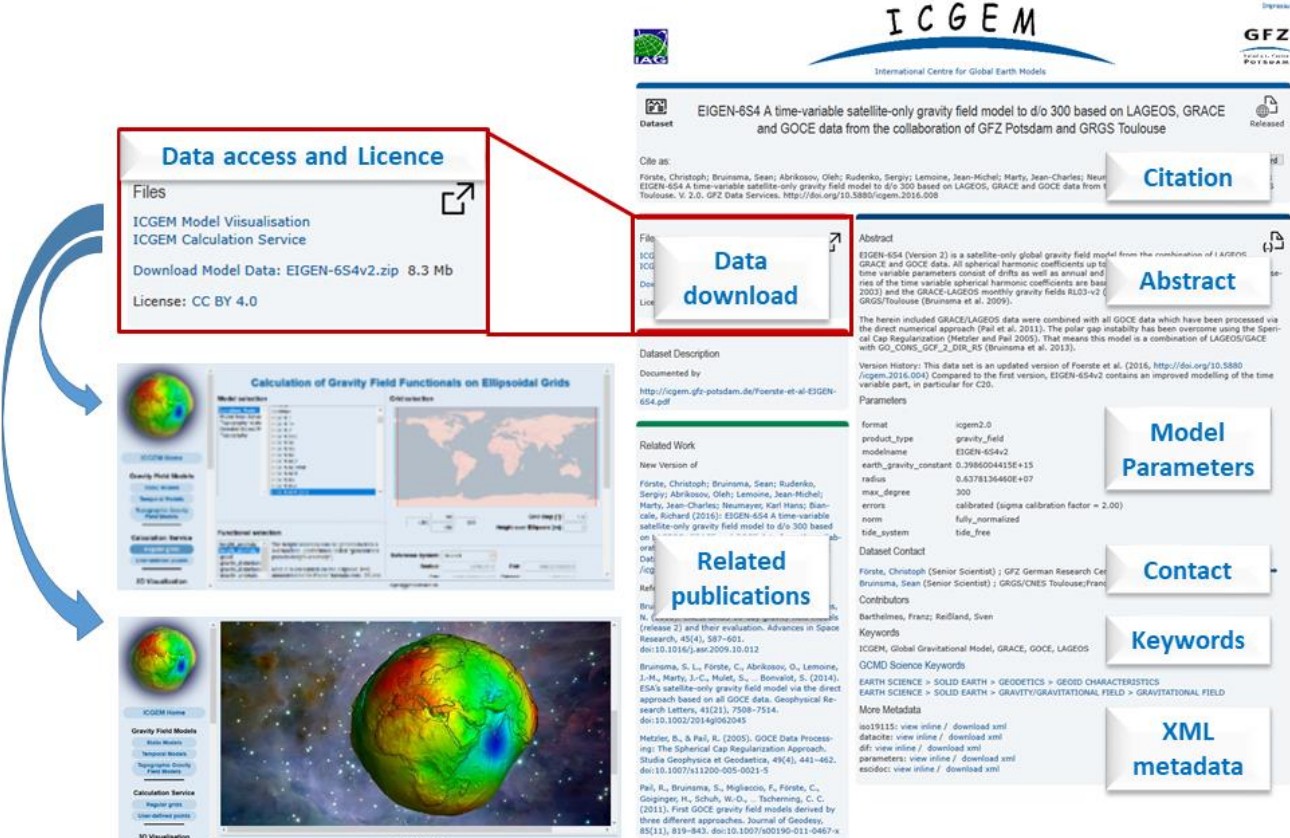

**Figure 19: Example for and features of an ICGEM DOI Landing Page (EIGEN-6S4, http://doi.org/10.5880/icgem.2016.008). The DOI Landing Page to the right is the visualisation of the metadata for data discovery that is collected for the DOI assignment. In addition to the links to the data, it includes the citation information, a data description (abstract), a table of model parameters, contact information, keywords and links to related publications. The metadata is also provided in machine-readable XML format. The key elements of the DOI metadata is also included in the header of the data files (model coefficients). The left part of the figure is a close-up of the data download section. GFZ Data Services stores a copy of the model data and provides direct links to the ICGEM Visualisation and Calculation Services with the pre-selection of the actual model.**



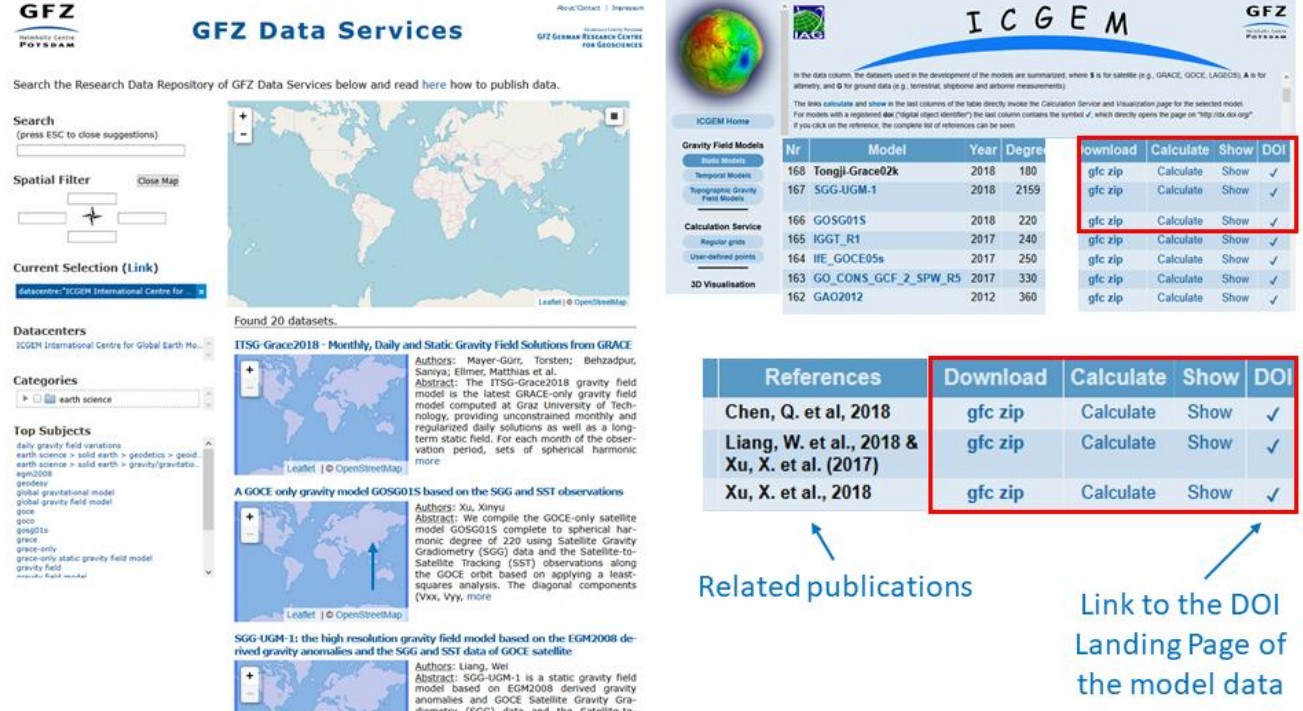

**Figure 20: Illustration of the two different possibilities to access the DOI-referenced ICGEM models: via the Catalogue of GFZ Data Services (left) and the ICGEM Website (right). GFZ Data Services have created a specific data centre for the ICGEM Service (http://dataservices.gfz-potsdam.de/portal/?fq=datacentre_facet:%22DOIDB.ICGEM%20-%20ICGEM%20International%20Centre%20for%20Global%20Earth%20Models%22). On the ICGEM Website (to the right, table of models), each ICGEM model with assigned DOI is linked to the DOI Landing page via the mark in the DOI column.**

## 4 Documentation

The documentation section of the ICGEM Service consists of five subsections: Frequently asked questions, theory, references,
10  latest changes, and discussion forum. The ICGEM team responds to users' questions as soon as possible in the discussion forum. During the last few years, there were common questions from advanced users, researchers and students that are fundamental to do thorough analyses in different applications. The ICGEM team has collected frequently asked questions (FAQs) selected from the users' and provides this collection with answers as a pdf document for the users. The questions are answered to meet the needs of both users from different scientific disciplines and industry related background, and experts in
15  the field of physical geodesy. The FAQs list is regularly updated when new questions accumulate. The last version of the FAQs can be accessed via the following web address http://icgem.gfz-potsdam.de/faq.

Although the theory of the global gravity field modelling and the calculations of gravity field functionals are not presented in this paper, it is most fundamental to the development of the ICGEM Service and a detailed documentation is reported in



Barthelmes, 2013. This scientific technical report includes the potential theory and approximations that are used in the global gravity field modelling. The formulas used in the computation of the gravity field functional are essential for the ICGEM calculation service and detailed in the GFZ scientific report.

ICGEM does not only collect gravitational model, but also pay attention on the full documentation of the models. New model releases, new documentation, conference and symposium presentations can be found in the ICGEM Home page and in the list of latest changes. For the convenience of the users, all relevant sources are listed in this reference list. This will ensure that the service and its components are available at the same place.

Moreover, the ICGEM Service provides a gravity field discussion forum (http://icgem.gfz-potsdam.de/guestbook) which provides users with a platform to communicate with the ICGEM team and other scientists working on similar topics. Apart from fulfilling the requirements of the service, this platform has also been used as a tool for educational purposes in which undergraduate or graduate students communicate with the ICGEM team directly. Since the interaction between the users and ICGEM team members has evolved and included extensive communications including e-mails, the definition of the old 'guest book' was redefined and has been modified into the forum in 2016 which better represents the current status of the platform.

The new version of the forum should give the users the opportunity to discuss any topic related to gravity field among themselves or answer each other's question and probably share data in the future. In the following years, we propose to establish sub-sections for different topics and expand the discussion forum to be unique in this field. Anyone without any registration requirement should still be able to write comments in the forum which will be publicly available after approval of the ICGEM team.

## 5 Web Programming

The original ICGEM website was established in 2003 and was based on plain HTML, Java Applets and Perl scripts. In 2016, several components of the website received an upgrade. The webpages are generated with the Python based framework CherryPy and Jinja2 templates. A MongoDB serves as database backend (see Figure 21 for an overview of the Web programming scheme). In addition, the 3D Visualization received minor changes and the user interface of the Calculation Service has been re-designed. The programs used in the calculation are the same as providing identical results and make use of the database to display the status of individual calculations.

***Visualisation Service***: The Visualization Service received minor updates primarily to ease the maintenance. The information of all models is kept in the MongoDB database. To visualize a specific model in user's browser, the ICGEM server creates a scene, dynamically applies changes to this scene and returns the resulting picture for display. Changes to the scene could be the rotation of the visualized model or changes of parameters.

***Table(s) of Models:*** This page simply requests all the model information from the MongoDB and displays it in a table, if a user wants to download a model, the model file is requested from the Model storage.

*Discussion Forum:* Posts from users are saved in the MongoDB and those approved by the ICGEM team are displayed on the page.

*Calculation Service (Browser):* When the user clicks on "Start Calculation", the browser sends the calculation settings to the server. The calculation settings are stored in the MongoDB database and the user is redirected to the calculation status page. The calculation progress is periodically updated from the MongoDB database. When the calculation is done, the results are displayed and download links are shown.

*Calculation Service:* The Calculation Service is a dedicated program on the server that requests new calculations from the MongoDB. For new calculations a calculation process is started and progress is stored in the MongoDB, to be requested by the calculation status page. After the calculation is done, the resulting grid is converted to an image. The number of concurrent calculation processes is limited, so a new calculation will have to wait until a slot is free.

*Evaluation (Browser):* For each model, the GNSS/levelling information is saved in the MongoDB. For the spectral domain evaluation, additionally, the images of the evaluation results are loaded and displayed.

*Evaluation Service:* In the MongoDB one model is selected as reference model, all other models have an attribute indicating which model was used for the evaluation. The Evaluation Service runs in regular intervals and requests the evaluation status of all models from the database. Models, which were not evaluated with the current reference model, will be automatically re-evaluated. When a new model is added, it will automatically be evaluated within some minutes. If the reference model changes, all models will be reevaluated.

*Static Sites:* The Static Sites like the FAQ and Theory are only a Jinja2 template and do not need any information from the MongoDB.

The overall architecture of the ICGEM website and its services is visualized in Figure 21. Pivotal component is the MongoDB database. On the one hand, this database is used to feed forms with information about models or discussion posts and to link to file storages, and on the other hand, the MongoDB database decouples most of the server components from the users internet browser. Therefore, long-lasting model calculations can be bookmarked for later inspection of the results. Because of its dynamic behavior, the Visualization Service requires direct communication with the users internet browser and the MongoDB database is not used to exchange information.





**Figure 21: Illustration of the scheme of the ICGEM Website in terms of its web programming. Arrows indicate the direction of data flows. Cylinders are the default symbol for Databases and sideway cylinders are used for file-based data stores. Model Storage is file based, the Calculation Results are the calculated Grid files and the images converted from them. Evaluation Results are only the generated images on the Spectral domain page, the values in the GNSS/levelling page derive from the MongoDB. At the bottom of the scheme, every grey-white rectangle represents a subpage on the Website. Similar pages, like the different Visualization pages or the Tables of Models, are presented together, and we included only the static pages, that are more interesting from a technical perspective.**



## 6 Conclusions and Future Aspects

The ICGEM Service is a worldwide used service, which continues to maintain its content and develop new features based on users' needs and requests since its establishment. Over the years, ICGEM has become the unique platform for providing comprehensive access to static, temporal, and topographic gravity field models and their documentation as well as to the online
calculation and visualisation services of the gravity field functionals.

In the near future, the old G3 Browser, which showed the time variation of gravity field at any desired point or pre-defined basin, will be available again with improved features developed for both advanced researchers and educational purposes. A specific web interface will be made available for the user to calculate and visualise the time series of mass variations. The results again will be available both in .png and. ASCII formats.

The other contribution will focus on the collection and provision of the $C_{20}$ coefficient time series from different institutions. This feature has been requested by advanced users who indicated that there is a need to collect these values from all developers in a common platform enabling access for the scientists and use for different purposes.

The discussion forum will be divided into sub sections and formatted for the user to communicate the other users as well as the ICGEM team and find answers to possible questions. If requested by the users, data sharing such as terrestrial gravity
measurements and GNSS/levelling derived geoid undulations and exchange can also be developed under the ICGEM web settings safely. Moreover, creation of an e-mail subscription list for the delivery monthly updates to the interested users is under discussion. These are possible options and opportunities to share the science and its products.

## 7 Data availability

The website of ICGEM with all model data, documentation and services as described in this article is available via
http://icgem.gfz-potsdam.de/. In addition, all gravity models with assigned DOI are also accessible via GFZ Data Services catalogue (http://dataservices.gfz-potsdam.de/portal/?fq=datacentre_facet:%22DOIDB.ICGEM%20-%20ICGEM%20International%20Centre%20for%20Global%20Earth%20Models%22). As the purpose of this article is the description of the ICGEM Service with all its features, including the DOI Service, we do not provide an additional DOI to all model data but refer the users to directly access the data via the ICGEM Website or GFZ Data Services.



*Author contributions:* ESI is responsible for the maintenance and the development of the ICGEM Service as of January 2018, prepared and coordinated the manuscript, FB developed the software and tools used in the ICGEM Service and contributed in the preparation and review of the manuscript, SR is responsible for the maintenance of the Service together with ESI, does the web programming of the ICGEM Service and contributed to the preparation of Section 5, KE deals with the DOI assignment

5    to the models in the Service, prepared subsection 3.5 and reviewed the paper and suggested improvements, CF reviewed the paper content and suggested improvements, FF reviewed the paper content and suggested improvements, HS reviewed the IAG Service and IUGG relevant sections and suggested improvements.

*Competing interest:* The authors declare that they have no conflict of interest.

**Acknowledgements**

We would like to acknowledge the past and present scientists who actively worked on the development of the ICGEM Service as well as helped in its establishment. ICGEM would not be in its current stage without the contribution of Wolfgang Köhler. We would like to acknowledge his contributions in the development of 3D visualisation software since the beginning of the

15    service until 2017. The founding fathers of the ICGEM Service are Christoph Reigber and Peter Schwintzer (deceased 2005) who also established the funding for the ICGEM Service. Christoph Dahle is acknowledged for his contributions in the temporal gravity field related components and Svetozar Petrovic for the delightful discussions during the evolution of ICGEM Website. Christian Voigt, Damian Ulbricht and Wouter van der Wal are thanked for reviewing a previous version of the paper. Lastly, as mentioned throughout the manuscript, user-response, questions and discussions with the ICGEM team significantly

20    helped and are still helping to develop, improve and further develop the tools. Therefore, we would like to especially acknowledge the user community contributing to the ICGEM's development with their questions, recommendations and feedback. Finally, Generic Mapping Tools (GMT) is acknowledged. The ICGEM Service would not be in its current shape without the advanced and sophisticated GMT mapping tools.



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

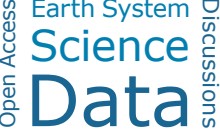

**Appendix 1: Other features of the calculation service**

A- Different projections are introduced in the background script which uses GMT version 5.0. The figures are adjusted based on the calculation area. The below figures show geoid undulations and height anomalies in a) North Hemisphere above latitude 50 degrees b) South America.

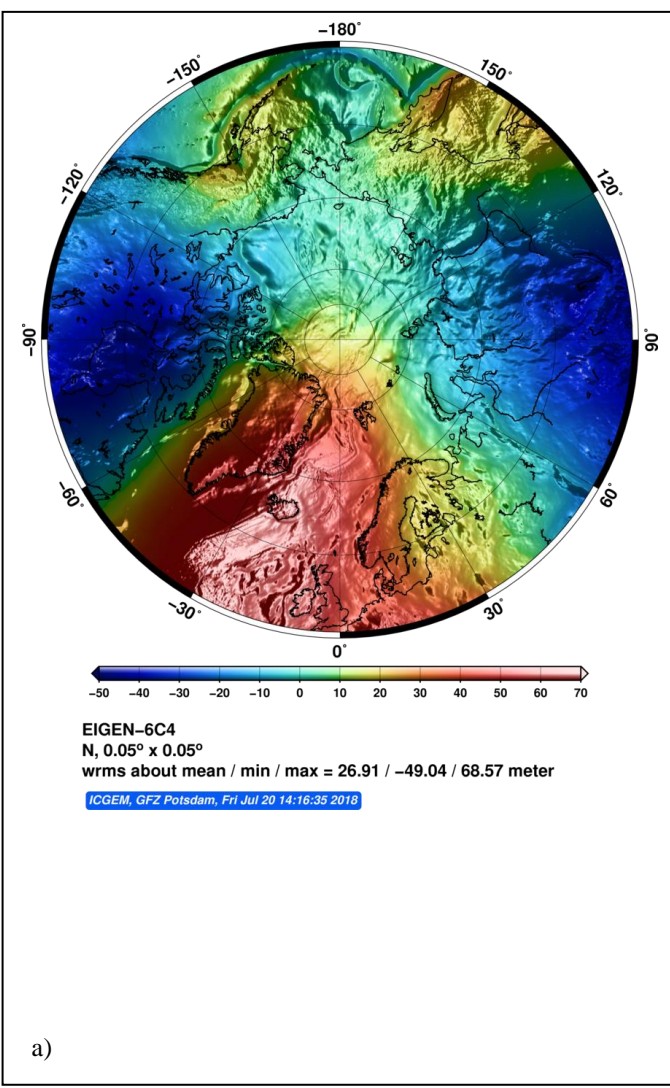

a)

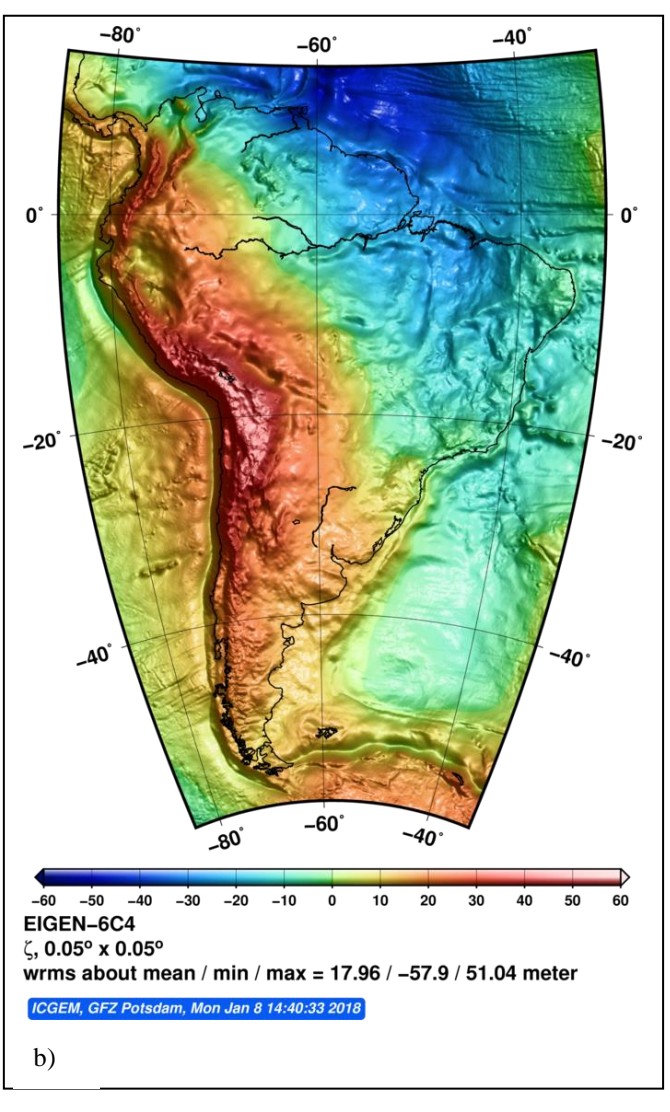

b)



B- Cross section calculation of gravity field functionals along latitude. The below figures show cross section along latitude of a) gravity of a potential geostationary satellite at an altitude of approximately 36000 km b) gravity on the earth surface along the equator.

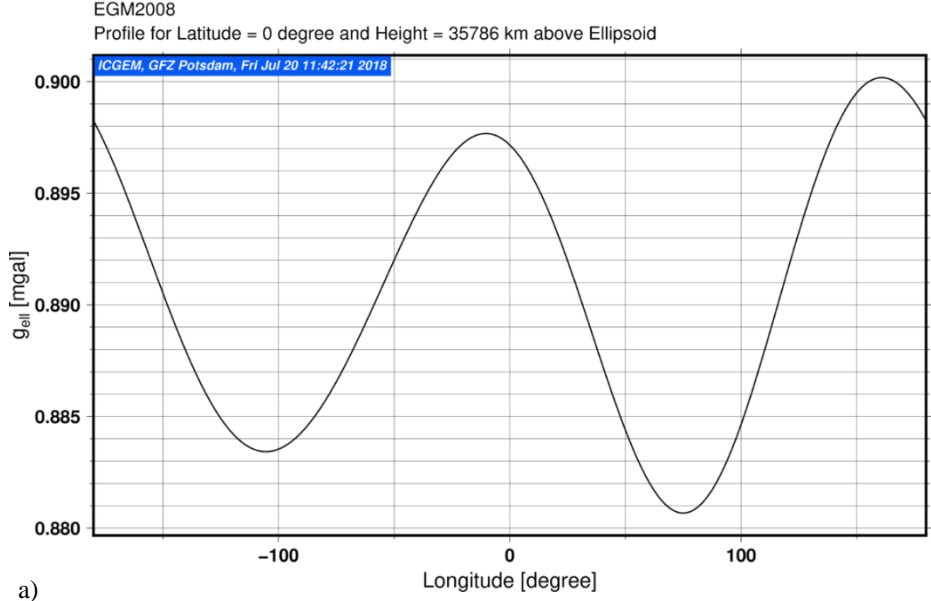

a)

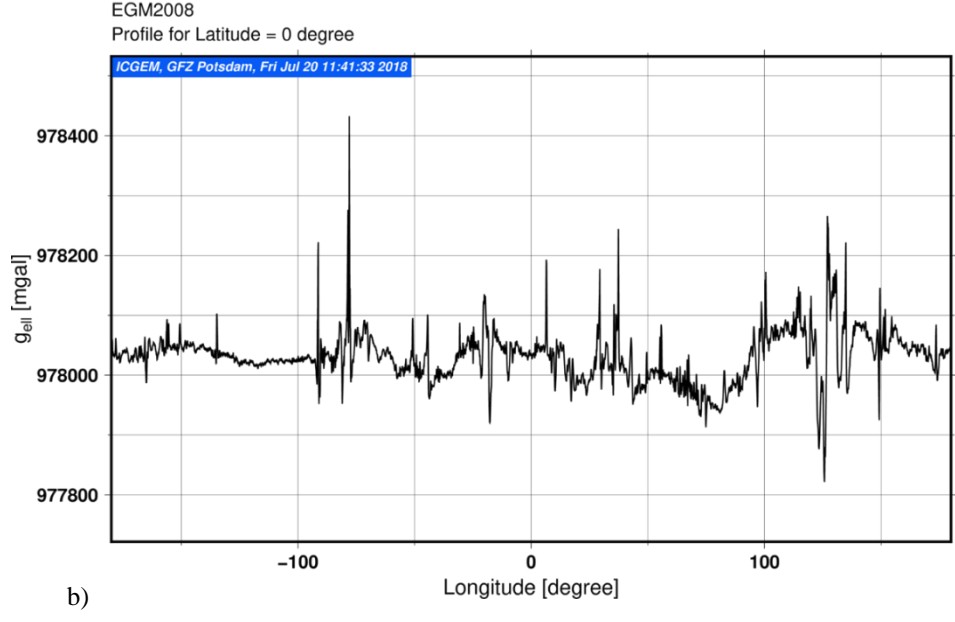

b)



Earth System
Science
Data

C- Cross section calculation of the gravity field functionals along longitude. The figures below show a) Classical gravity anomalies at grid points where the red line indicated the cross section along longitude b) Cross section classical gravity anomalies along longitude c) Cross section geoid undulations along longitude

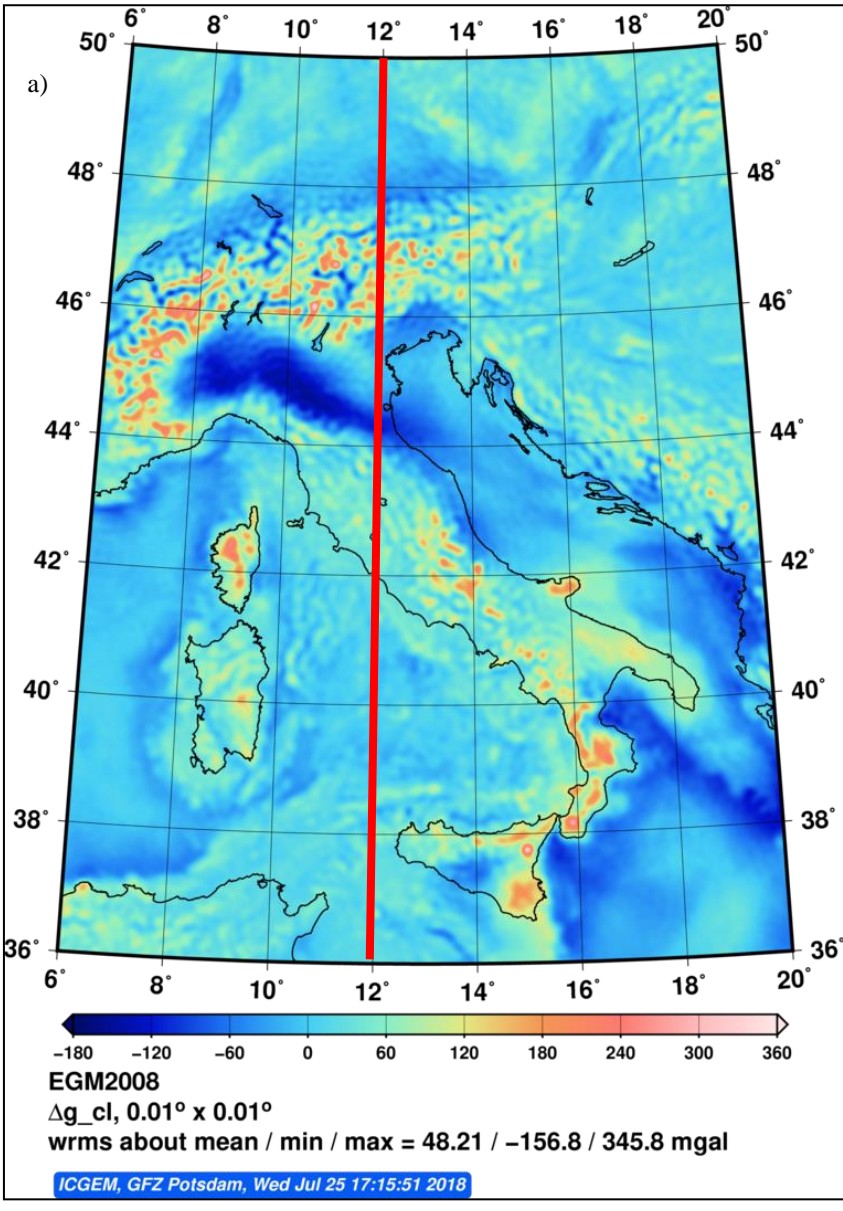



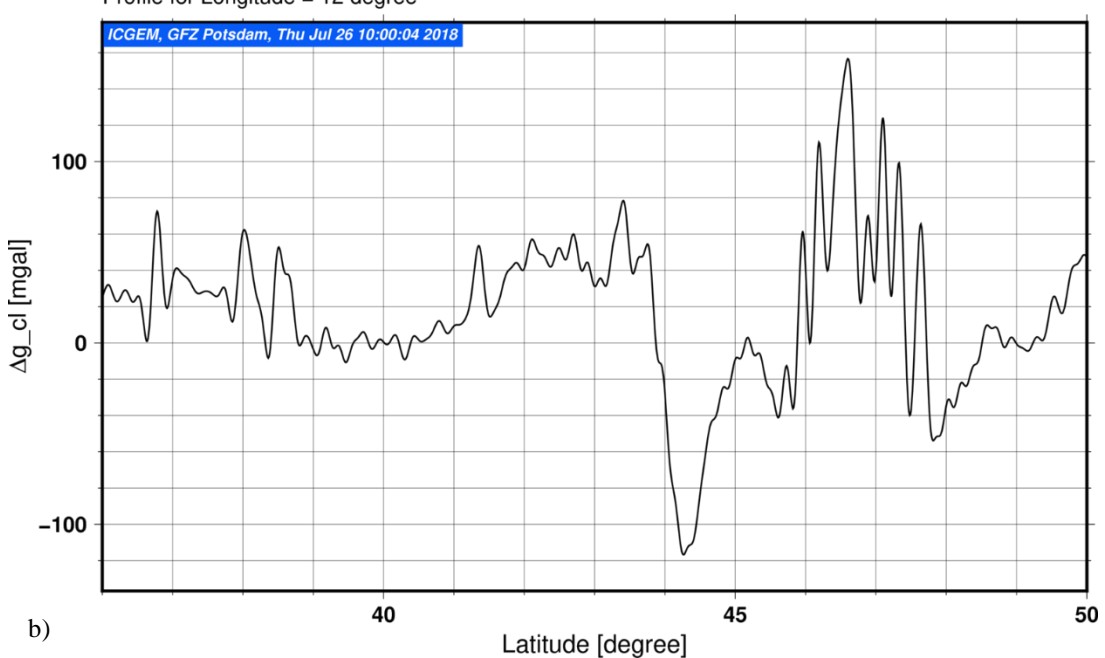

b)

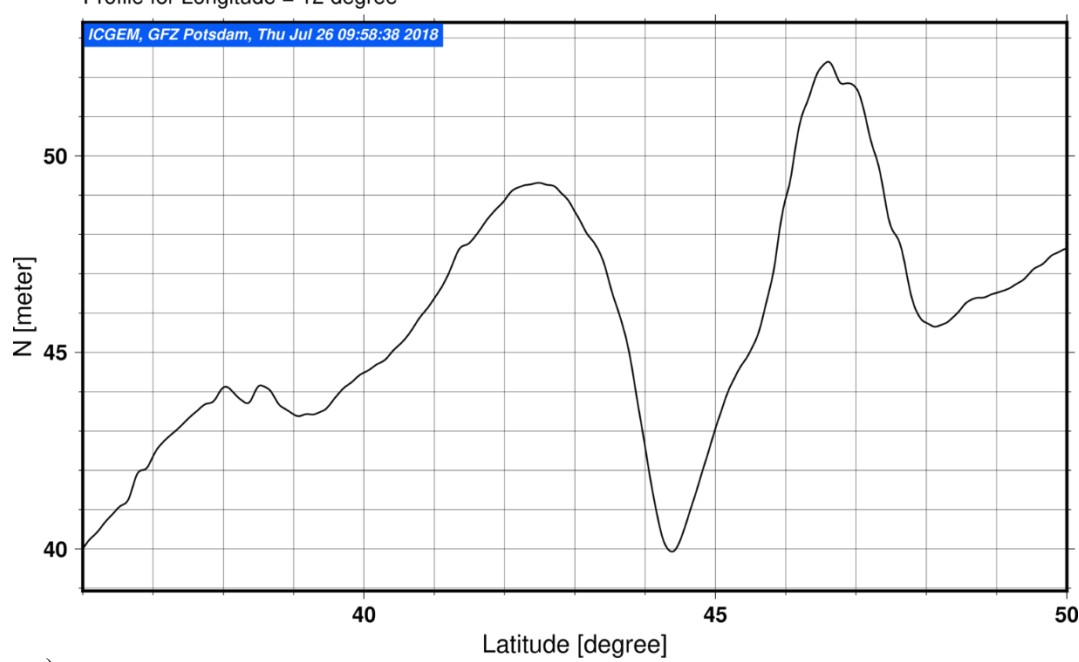

c)



D- Visualisation of other celestial bodies. Figures below show geoid undulations of  a) Moon b) Mars c) Venus

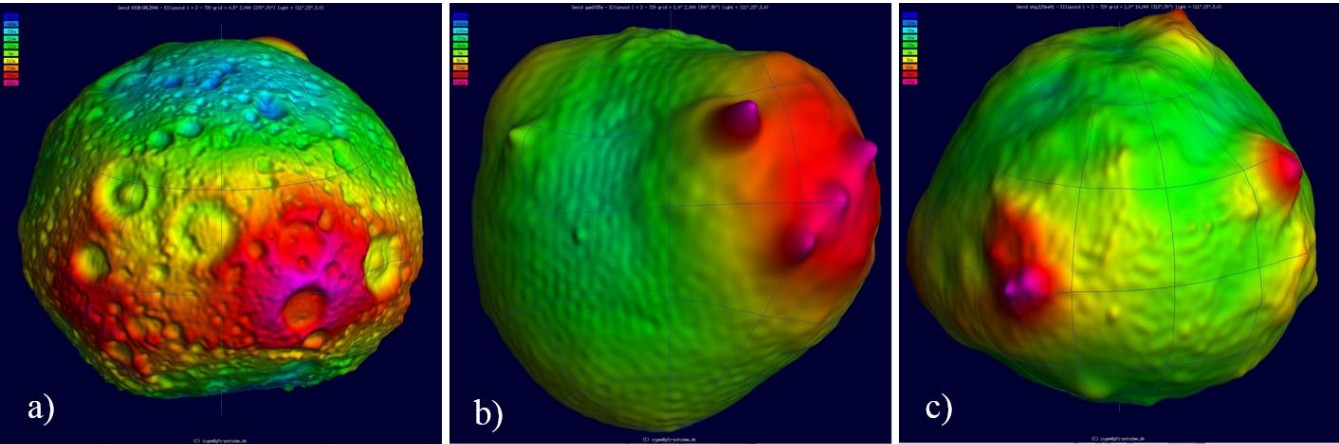



**Appendix 2: List of static gravity field modes available in ICGEM service (Status 01/12/2018)**

The six column represents: the number of the model in the ICGEM database, name of the model, release year of the model, maximum spherical harmonic degree expansion of the model, data sources used in the model and the references for the model development and description, respectively.

| | | | | | |
|---|---|---|---|---|---|
| *168* | **Tongji-Grace02k** | **2018** | **180** | **S(GRACE)** | **Chen, Q. et al, 2018** |
| *167* | SGG-UGM-1 | 2018 | 2159 | EGM2008, S(GOCE) | **Liang, W. et al., 2018 & Xu, X. et al. (2017)** |
| *166* | GOSG01S | 2018 | 220 | S(GOCE) | **Xu, X. et al., 2018** |
| *165* | IGGT_R1 | 2017 | 240 | S(GOCE) | **Lu, B. et al, 2017** |
| *164* | IfE_GOCE05s | 2017 | 250 | S(GOCE) | **Wu, H. et al, 2017** |
| *163* | GO_CONS_GCF_2_SPW_R5 | 2017 | 330 | S(GOCE) | **Gatti, A. et al, 2016** |
| *162* | GAO2012 | 2012 | 360 | A, G, S(GOCE), S(GRACE) | **Demianov, G. et al, 2012** |
| *161* | XGM2016 | 2017 | 719 | A, G, S(GOCO05s) | **Pail, R. et al, 2017** |
| *160* | **Tongji-Grace02s** | 2017 | 180 | S(Grace) | **Chen, Q. et al, 2016** |
| *159* | **NULP-02s** | 2017 | 250 | S(Goce) | **A.N. Marchenko et al, 2016** |
| *158* | **HUST-Grace2016s** | 2016 | 160 | S(Grace) | **Zhou, H. et al, 2016** |
| *157* | ITU_GRACE16 | 2016 | 180 | S(Grace) | **Akyilmaz, O. et al, 2016** |
| *156* | ITU_GGC16 | 2016 | 280 | S(Goce), S(Grace) | **Akyilmaz, O. et al, 2016** |
| *155* | EIGEN-6S4 (v2) | 2016 | 300 | S(Goce), S(Grace), S(Lageos) | **Förste, C. and Bruinsma, S.L., 2016** |
| *154* | GOCO05c | 2016 | 720 | (see model), A, G, S | **Fecher, T. et al, 2016** |
| *153* | GGM05C | 2015 | 360 | A, G, S(Goce), S(Grace) | **Ries, J. et al, 2016** |
| *152* | **GECO** | 2015 | 2190 | EGM2008, S(Goce) | **Gilardoni, M. et al, 2016** |
| *151* | GGM05G | 2015 | 240 | S(Goce), S(Grace) | **Bettadpur, S. et al, 2015** |
| *150* | GOCO05s | 2015 | 280 | (see model), S | **Mayer-Gürr, T. et al, 2015** |
| *149* | GO_CONS_GCF_2_SPW_R4 | 2014 | 280 | S(Goce) | **Gatti, A. et al, 2014** |
| *148* | EIGEN-6C4 | 2014 | 2190 | A, G, S(Goce), S(Grace), S(Lageos) | **Förste, Christoph et al, 2014** |
| *147* | ITSG-Grace2014s | 2014 | 200 | S(Goce) | **Mayer-Gürr, T. et al, 2014** |
| *146* | ITSG-Grace2014k | 2014 | 200 | S(Grace) | **Mayer-Gürr, T. et al, 2014** |
| *145* | GO_CONS_GCF_2_TIM_R5 | 2014 | 280 | S(Goce) | **Brockmann, J. M. et al, 2014** |
| *144* | GO_CONS_GCF_2_DIR_R5 | 2014 | 300 | S(Goce), S(Grace), S(Lageos) | **Bruinsma, S. L. et al, 2013** |
| *143* | JYY_GOCE04S | 2014 | 230 | S(Goce) | **Yi, Weiyong et al, 2013** |
| *142* | GOGRA04S | 2014 | 230 | S(Goce), S(Grace) | **Yi, Weiyong et al, 2013** |
| *141* | EIGEN-6S2 | 2014 | 260 | S(Goce), S(Grace), S(Lageos) | **Rudenko, Sergei et al, 2014** |
| *140* | GGM05S | 2014 | 180 | S(Grace) | **Tapley, B.D. et al, 2013** |
| *139* | EIGEN-6C3stat | 2014 | 1949 | A, G, S(Goce), S(Grace), S(Lageos) | **Förste, C. et al, 2012** |
| *138* | Tongji-GRACE01 | 2013 | 160 | S(Grace) | **Shen, Y. et al, 2013** |
| *137* | JYY_GOCE02S | 2013 | 230 | S(Goce) | **Yi, Weiyong et al, 2013** |
| *136* | GOGRA02S | 2013 | 230 | S(Goce), S(Grace) | **Yi, Weiyong et al, 2013** |
| *135* | **ULux_CHAMP2013s** | 2013 | 120 | S(Champ) | **Weigelt, M. et al, 2013** |
| *134* | ITG-Goce02 | 2013 | 240 | S(Goce) | **Schall, Judith et al, 2014** |
| *133* | GO_CONS_GCF_2_TIM_R4 | 2013 | 250 | S(Goce) | **Pail, Roland et al, 2011** |
| *132* | GO_CONS_GCF_2_DIR_R4 | 2013 | 260 | S(Goce), S(Grace), S(Lageos) | **Bruinsma, S. L. et al, 2013** |
| *131* | EIGEN-6C2 | 2012 | 1949 | A, G, S(Goce), S(Grace), S(Lageos) | **Förste, C. et al, 2012** |
| *130* | DGM-1S | 2012 | 250 | S(Goce), S(Grace) | **Farahani, H. Hashemi et al, 2013** |
| *129* | GOCO03s | 2012 | 250 | S(Goce), S(Grace) | **Mayer-Gürr, T. et al, 2012** |
| *128* | GO_CONS_GCF_2_DIR_R3 | 2011 | 240 | S(Goce), S(Grace), S(Lageos) | **Bruinsma, S.L. et al, 2010** |
| *127* | GO_CONS_GCF_2_TIM_R3 | 2011 | 250 | S(Goce) | **Pail, R. et al, 2010** |
| *126* | GIF48 | 2011 | 360 | A, G, S(Grace) | **Ries, J.C. et al, 2011** |
| *125* | EIGEN-6C | 2011 | 1420 | A, G, S(Goce), S(Grace), S(Lageos) | **Förste, C. et al, 2011** |
| *124* | EIGEN-6S | 2011 | 240 | S(Goce), S(Grace), S(Lageos) | **Förste, C. et al, 2011** |
| *123* | GOCO02s | 2011 | 250 | S(Goce), S(Grace) | **Goiginger, H. et al, 2011** |
| *122* | AIUB-GRACE03S | 2011 | 160 | S(Grace) | **Jäggi, A. et al,** |



| | | | | | |
|---|---|---|---|---|---|
| 121 | GO_CONS_GCF_2_DIR_R2 | 2011 | 240 | S(Goce) | Bruinsma, S.L. et al, 2010 |
| 120 | GO_CONS_GCF_2_TIM_R2 | 2011 | 250 | S(Goce) | Pail, Roland et al, 2011 |
| 119 | GO_CONS_GCF_2_SPW_R2 | 2011 | 240 | S(Goce) | Migliaccio, F. et al, 2011 |
| 118 | GO_CONS_GCF_2_DIR_R1 | 2010 | 240 | S(Goce) | Bruinsma, S.L. et al, 2010 |
| 117 | GO_CONS_GCF_2_TIM_R1 | 2010 | 224 | S(Goce) | Pail, R. et al, 2010 |
| 116 | GO_CONS_GCF_2_SPW_R1 | 2010 | 210 | S(Goce) | Migliaccio, F. et al, 2010 |
| 115 | GOCO01S | 2010 | 224 | S(Champ), S(Grace) | Pail, R. et al, 2010 |
| 114 | EIGEN-51C | 2010 | 359 | A, G, S(Champ), S(Grace) | Bruinsma, S.L. et al, 2010 |
| 113 | AIUB-CHAMP03S | 2010 | 100 | S(Champ) | Prange, L., 2010 |
| 112 | EIGEN-CHAMP05S | 2010 | 150 | S(Champ) | Flechtner, Frank et al, 2010 |
| 111 | ITG-Grace2010s | 2010 | 180 | S(Grace) | Mayer-Gürr, T. et al, 2010 |
| 110 | AIUB-GRACE02S | 2009 | 150 | S(Grace) | Jäggi, A. et al, 2012 |
| 109 | GGM03C | 2009 | 360 | A, G, S(Grace) | Tapley, B.D. et al, 2007 |
| 108 | GGM03S | 2008 | 180 | S(Grace) | Tapley, B.D. et al, 2007 |
| 107 | AIUB-GRACE01S | 2008 | 120 | S(Grace) | Jäggi, A. et al, 2010 |
| 106 | EIGEN-5S | 2008 | 150 | S(Grace), S(Lageos) | Förste, C. et al, 2008 |
| 105 | EIGEN-5C | 2008 | 360 | A, G, S(Grace), S(Lageos) | Förste, C. et al, 2008 |
| 104 | EGM2008 | 2008 | 2190 | A, G, S(Grace) | Pavlis, N.K. et al, 2008 |
| 103 | ITG-Grace03 | 2007 | 180 | S(Grace) | Mayer-Gürr, T. et al, 2007 |
| 102 | AIUB-CHAMP01S | 2007 | 70 | S(Champ) | Prange, L. et al, 2009 |
| 101 | ITG-Grace02s | 2006 | 170 | S(Grace) | Mayer-Gürr, T. et al, 2006 |
| 100 | EIGEN-GL04S1 | 2006 | 150 | S(Grace), S(Lageos) | Förste, C. et al, 2006 |
| 99 | EIGEN-GL04C | 2006 | 360 | A, G, S(Grace), S(Lageos) | Förste, C. et al, 2006 |
| 98 | eigen-cg03c | 2005 | 360 | A, G, S(Champ), S(Grace) | Förste, C. et al, 2005 |
| 97 | GGM02C | 2004 | 200 | A, G, S(Grace) | Tapley, B. et al, 2005 |
| 96 | GGM02S | 2004 | 160 | S(Grace) | Tapley, B. et al, 2005 |
| 95 | EIGEN-CG01C | 2004 | 360 | A, G, S(Champ), S(Grace) | Reigber, C. et al, 2006 |
| 94 | EIGEN-CHAMP03S | 2004 | 140 | S(Champ) | Reigber, C. et al, 2004 |
| 93 | EIGEN-GRACE02S | 2004 | 150 | S(Grace) | Reigber, Christoph et al, 2005 |
| 92 | TUM-2S | 2004 | 60 | S(Champ) | Wermuth, M. et al, 2004 |
| 91 | DEOS_CHAMP-01C | 2004 | 70 | S(Champ) | Ditmar, P. et al, 2005 |
| 90 | ITG_Champ01K | 2003 | 70 | S(Champ) | Ilk, K.H. et al, 2005 |
| 89 | ITG_Champ01S | 2003 | 70 | S(Champ) | Ilk, K.H. et al, 2005 |
| 88 | ITG_Champ01E | 2003 | 75 | S(Champ) | Ilk, K.H. et al, 2005 |
| 87 | TUM-2Sp | 2003 | 60 | S(Champ) | Földváry, Lóránt et al, 2005 |
| 86 | TUM-1S | 2003 | 60 | S(Champ) | Gerlach, Ch et al, 2003 |
| 85 | GGM01C | 2003 | 200 | S(Grace), TEG4 | Tapley, B. D. et al, 2004 |
| 84 | GGM01S | 2003 | 120 | S(Grace) | Tapley, B. D. et al, 2003 |
| 83 | EIGEN-GRACE01S | 2003 | 140 | S(Grace) | Reigber, C. et al, 2003 |
| 82 | EIGEN-CHAMP03Sp | 2003 | 140 | S(Champ) | Reigber, C. et al, 2004 |
| 81 | EIGEN-2 | 2003 | 140 | S(Champ) | Reigber, C. et al, 2003 |
| 80 | EIGEN-1 | 2002 | 119 | S(Champ) | Reigber, C. et al, 2003 |
| 79 | EIGEN-1s | 2002 | 119 | GRIM5, S | Reigber, Christoph et al, 2002 |
| 78 | PGM2000a | 2000 | 360 | A, G, S | Pavlis, N.K. et al, 2000 |
| 77 | TEG4 | 2000 | 200 | A, G, S | Tapley, B.D. et al, 2000 |
| 76 | GRIM5c1 | 1999 | 120 | A, G, S | Gruber, T. et al, 1999 |
| 75 | GRIM5s1 | 1999 | 99 | S | Biancale, R. et al, 2000 |
| 74 | GRIM4S4G | 1999 | 70 | GRIM4S4, S(GFZ-1) | König, R. et al, 1999 |
| 73 | GFZ97 | 1997 | 359 | A, G, PGM062w | Gruber, T. et al, 1997 |
| 72 | EGM96 | 1996 | 360 | A, EGM96S, G | Lemoine, F.G. et al, 1998 |
| 71 | GFZ96 | 1996 | 359 | A, G, PGM055 | Gruber, Th. et al, 1996 |
| 70 | TEG3 | 1996 | 70 | A, G, S | Tapley, B.D. et al, 1997 |
| 69 | EGM96s | 1996 | 70 | S | Lemoine, F.G. et al, 1998 |
| 68 | GFZ95A | 1995 | 360 | A, G, GRIM4C4 | Gruber, Thomas et al, 1996 |
| 67 | GRIM4c4 | 1995 | 72 | A, G, S | Schwintzer, P. et al, 1997 |
| 66 | GRIM4s4 | 1995 | 70 | S | Schwintzer, P. et al, 1997 |



| 65 | JGM3 | 1994 | 70 | A, G, S | Tapley, B. D. et al, 1996 |
|----|------|------|----|---------|---------------------------|
| 64 | JGM2 | 1994 | 70 | A, G, S | Nerem, R. S. et al, 1994 |
| 63 | JGM2s | 1994 | 70 | S | Nerem, R. S. et al, 1994 |
| 62 | GFZ93B | 1993 | 360 | A, G, GRIM4C3 | Gruber, T. et al, 1993 |
| 61 | GFZ93a | 1993 | 360 | A, G, GRIM4C3 | Gruber, T. et al, 1993 |
| 60 | JGM1 | 1993 | 70 | A, G, S | Nerem, R. S. et al, 1994 |
| 59 | JGM1s | 1993 | 60 | S | Nerem, R. S. et al, 1994 |
| 58 | OGE12 | 1992 | 360 | A, G, GRIM4C2 | Gruber, T. et al, 1992 |
| 57 | GRIM4C3 | 1992 | 60 | A, G, S | Schwintzer, P. et al, 1993 |
| 56 | GRIM4s3 | 1992 | 60 | S | Schwintzer, P. et al, 1993 |
| 55 | OSU91a | 1991 | 360 | A, G, GEMT2 | Rapp, R.H. et al, 1991 |
| 54 | GRIM4c2 | 1991 | 50 | A, G, S | Schwintzer, P. et al, 1992 |
| 53 | GRIM4s2 | 1991 | 50 | S | Schwintzer, P. et al, 1992 |
| 52 | GEMT3 | 1991 | 50 | A, G, S | Lerch, F.J. et al, 1992 |
| 51 | GEMT3s | 1991 | 50 | S | Lerch, F.J. et al, 1992 |
| 50 | TEG2B | 1991 | 54 | A, G, S | Tapley, B.D. et al, 1991 |
| 49 | TEG2 | 1990 | 54 | A, G, S | Tapley, B.D. et al, 1991 |
| 48 | GRIM4c1 | 1990 | 50 | A, G, S | Schwintzer, P. et al, 1991 |
| 47 | GRIM4s1 | 1990 | 50 | S | Schwintzer, P. et al, 1991 |
| 46 | GEMT2 | 1989 | 50 | A, G, S | Marsh, J. G. et al, 1990 |
| 45 | GEMT2S | 1989 | 50 | S | Marsh, J. G. et al, 1990 |
| 44 | POEM-L1 | 1989 | 20 | S(Lageos) | Dietrich, R. et al, 1989 |
| 43 | TEG1 | 1988 | 50 | G, S | Tapley, B.D. et al, 1991 |
| 42 | OSU89b | 1989 | 360 | A, G, GEMT2 | Rapp, Richard H. and Pavlis, Nikolaos K., 1990 |
| 41 | OSU89a | 1989 | 360 | A, G, GEMT2 | Rapp, Richard H. and Pavlis, Nikolaos K., 1990 |
| 40 | GEMT1 | 1987 | 36 | S | Marsh, J. G. et al, 1988 |
| 39 | OSU86f | 1986 | 360 | A, G, GEML2 | Rapp, R.H. et al, 1986 |
| 38 | OSU86e | 1986 | 360 | A, G, GEML2 | Rapp, R.H. et al, 1986 |
| 37 | OSU86d | 1986 | 250 | A, G, GEML2 | Rapp, R.H. Cruz, J.Y. et al, 1986 |
| 36 | OSU86c | 1986 | 250 | A, G, GEML2 | Rapp, R.H. Cruz, J.Y. et al, 1986 |
| 35 | GPM2 | 1984 | 200 | A, G, GEML2 | Wenzel, H.G., 1985 |
| 34 | GRIM3L1 | 1984 | 36 | A, G, S | Reigber, C. et al, 1985 |
| 33 | HAJELA84 | 1983 | 250 | G | Hajela, D.P. et al, 1984 |
| 32 | GPM1 | 1983 | 200 | A, G, GEM9 | Wenzel, H.G., 1985 |
| 31 | GRIM3B | 1983 | 36 | A, G, S | Reigber, C. et al, 1983 |
| 30 | GEML2 | 1982 | 30 | S | Lerch, F.J. et al, 1983 |
| 29 | GRIM3 | 1981 | 36 | A, G, S | Reigber, C. et al, 1983 |
| 28 | OSU81 | 1981 | 180 | A, G, GEM9 | Rapp, R.H. et al, 1981 |
| 27 | GEM10C | 1981 | 180 | A, G, GEM10B | Lerch, Francis J. et al, 1981 |
| 26 | OSU78 | 1978 | 180 | A, G, GEM9 | Rapp, R.H. et al, 1978 |
| 25 | GEM10b | 1978 | 36 | A, GEM10 | Lerch, F.J. et al, 1978 |
| 24 | GEM10a | 1978 | 30 | A, GEM10 | Lerch, F.J. et al, 1978 |
| 23 | GEM10 | 1977 | 30 | G, S | Lerch, Fancis J. et al, 1979 |
| 22 | GEM9 | 1977 | 30 | S | Lerch, Fancis J. et al, 1979 |
| 21 | GRIM2 | 1976 | 23 | G, S | Balmino, G. et al, 1976 |
| 20 | GEM8 | 1976 | 25 | G, S | Wagner, C.A. et al, 1976 |
| 19 | GEM7 | 1976 | 16 | S | Wagner, C.A. et al, 1976 |
| 18 | HARMOGRAV | 1975 | 36 | G | Dimitrijevich, V., 1975 |
| 17 | GRIM1 | 1975 | 31 | S | Balmino, G. et al, 1976 |
| 16 | KOCH74 | 1974 | 15 | G, S | Koch, K.R., 1974 |
| 15 | GEM6 | 1974 | 16 | G, S | Lerch, F.J. et al, 1974 |
| 14 | GEM5 | 1974 | 12 | S | Lerch, F.J. et al, 1974 |
| 13 | OSU73 | 1973 | 20 | G, GEM3 | Rapp, R.H. et al, 1973 |
| 12 | SE3 | 1973 | 24 | G, S | Gaposchkin, E.M. and Smithsonian Astrophysical Observatory, 1973 |





| | | | | | |
|---|---|---|---|---|---|
| *11* | **WGS72** | 1972 | 28 | G, S | **Seppelin, T.O. and WGS Committee,, 1974** |
| *10* | **GEM4** | 1972 | 16 | G, S | **Lerch, F.J. et al, 1972** |
| *9* | **GEM3** | 1972 | 12 | S | **Lerch, F.J. et al, 1972** |
| *8* | **GEM2** | 1972 | 22 | G, S | **Lerch, F.J. et al, 1972** |
| *7* | **GEM1** | 1972 | 22 | S | **Lerch, F.J. et al, 1972** |
| *6* | **KOCH71** | 1971 | 11 | G, S | **Koch, Karl-Rudolf and Witte, Bertold U., 1971** |
| *5* | **KOCH70** | 1970 | 8 | G, S | **Koch, Karl-Rudolf and Morrison, Foster, 1970** |
| *4* | **SE2** | 1969 | 22 | G, S | **Gaposchkin, E.M. Lambeck, K., 1970** |
| *3* | **OSU68** | 1968 | 14 | G, S | **Rapp, Richard H., 1968** |
| *2* | **WGS66** | 1966 | 24 | G | **WGS Committee, 1966** |
| *1* | **SE1** | 1966 | 15 | S | **Lundquist, C.A. et al, 1966** |





**Appendix 3: List of the centres producing GRACE results**

The processing standards to generate the GRACE Level-2 products of CSR, GFZ and JPL are also available in the Document Section of the GRACE archives at GFZ ISDC or JPL PO.DAAC

| GRACE monthly solutions from the 3 processing centers CSR, GFZ and JPL | |
|---|---|
| CSR Release 05 | (UTCSR Level-2 Processing Standards Document, Rev 4.0 May 29, 2012) |
| GFZ Release 05 | (GFZ GRACE Level-2 Processing, Revised Edition, January 2013) |
| JPL Release 05 | (PL Level-2 Processing Standards Document, Release 05.1 November 3, 2014) |

| GRACE / CHAMP monthly solutions from other groups | |
|---|---|
| AIUB Release 02 | (more information can be found here) |
| CNES_GRGS | (GRACE solutions release 03; more information can be found here) |
| DMT-1 | (more information can be found here) |
| EGSIEM | (GRACE monthly combined solutions from the EGSIEM project, more information can be found here) |
| GEO-Q | |
| HUST-Grace2016 | (GRACE monthly solutions from the Huazhong University of Science and Technology, Wuhan, PR China) |
| IGG_RL01 | (GRACE monthly solutions from the Institute of Geodesy and Geophysics, Chinese Academy of Sciences, China) |
| ITG | (more information can be found here) |
| ITSG-Grace2014 | (GRACE monthly solutions from the ITSG, TU Graz; more information can be found here) |
| ITSG-Grace2016 | (GRACE monthly solutions from the ITSG, TU Graz; more information can be found here) |
| Tongji Release 01 | (GRACE monthly solutions from the Tongji University, Shanghai, PR China) |
| Tongji Release 02 new version | (GRACE monthly solutions from the Tongji University, Shanghai, PR China) |
| Tongji Release 02 old version | (GRACE monthly solutions from the Tongji University, Shanghai, PR China) |
| ULux | (CHAMP monthly solutions from the University of Luxembourg) |
| WHU RL01 | (GRACE monthly solutions from the GNSS Research Center of Wuhan University, PR China) |

| GRACE weekly solutions | |
|---|---|
| **GFZ Release 05** | **(GFZ GRACE Level-2 Processing, Revised Edition, January 2013)** |

| GRACE daily solutions | |
|---|---|
| **ITSG-Grace2014** | **(more information can be found here)** |
| ITSG-Grace2016 | (more information can be found here) |

| SLR monthly solutions | |
|---|---|
| SLR-only monthly solutions from AIUB | |

| Non-isotropic smoothing | |
|---|---|
| AIUB Release 02 | (more information can be found here) |
| CSR Release 05 | (UTCSR Level-2 Processing Standards Document, Rev 4.0 May 29, 2012) |
| GFZ Release 05 | (GFZ GRACE Level-2 Processing, Revised Edition, January 2013) |
| HUST-Grace2016 | (GRACE monthly solutions from the Huazhong University of Science and Technology, Wuhan, PR China) |
| ITSG-Grace2014 | (GRACE monthly solutions from the ITSG, TU Graz; more information can be found here) |
| ITSG-Grace2016 | (GRACE monthly solutions from the ITSG, TU Graz; more information can be found here) |
| JPL Release 05 | (PL Level-2 Processing Standards Document, Release 05.1 November 3, 2014) |
| Tongji Release 01 | (GRACE monthly solutions from the Tongji University, Shanghai, PR China) |
| Tongji Release 02 new version | (GRACE monthly solutions from the Tongji University, Shanghai, PR China) |
| Tongji Release 02 old version | (GRACE monthly solutions from the Tongji University, Shanghai, PR China) |





**Appendix 4: List of the topographic gravity field models**

| Nr | Model | Year | Degree | Data | References |
|----|-------|------|--------|------|-----------|
| 18 | dV_ELL_Earth2014_5480_plusGRS80 | 2017 | 5480 | Topography | Rexer et al., (2017), Rexer, M. (2017) |
| 17 | dV_ELL_Earth2014_5480 | 2017 | 5480 | Topography | Rexer et al., (2017), Rexer, M. (2017) |
| 16 | dV_ELL_Earth2014_plusGRS80 | 2016 | 2190 | Topography | Rexer et al., (2016) |
| 15 | dV_ELL_Earth2014 | 2016 | 2190 | Topography | Rexer et al., (2016) |
| 14 | dV_ELL_RET2014_plusGRS80 | 2016 | 2190 | Topography | Rexer et al., (2016) |
| 13 | dV_ELL_RET2014 | 2016 | 2190 | Topography | Rexer et al., (2016) |
| 12 | REQ_TOPO_2015_plusGRS80 | 2015 | 2190 | Topography | Grombein et al., (2016) |
| 11 | RWI_TOPO_2015_plusGRS80 | 2015 | 2190 | Topography | Grombein et al., (2016) |
| 10 | REQ_TOPO_2015 | 2015 | 2190 | Topography | Grombein et al., (2016) |
| 9 | RWI_TOPO_2015 | 2015 | 2190 | Topography | Grombein et al., (2016) |
| 8 | RWI_TOIS_2012_plusGRS80 | 2014 | 1800 | Isostasy, Topography | Grombein et al., (2014) |
| 7 | RWI_ISOS_2012_plusGRS80 | 2014 | 1800 | Isostasy | Grombein et al., (2014) |
| 6 | RWI_TOPO_2012_plusGRS80 | 2014 | 1800 | Topography | Grombein et al., (2014) |
| 5 | RWI_TOIS_2012 | 2014 | 1800 | Isostasy, Topography | Grombein et al., (2014) |
| 4 | RWI_ISOS_2012 | 2014 | 1800 | Isostasy | Grombein et al., (2014) |
| 3 | RWI_TOPO_2012 | 2014 | 1800 | Topography | Grombein et al., (2014) |
| 2 | dV_ELL_RET2012_plusGRS80 | 2014 | 2190 | Topography | Claessens, S.J. and C. Hirt (2013) |
| 1 | dV_ELL_RET2012 | 2014 | 2190 | Topography | Claessens, S.J. and C. Hirt (2013) |





## Appendix 5: List of the gravity field models of other celestial bodies

| Object | Model | Year | Degree | References |
|--------|-------|------|--------|------------|
| *Mars* | ggm1025a | 2002 | 80 | **F.G. Lemoine et al, 2001** |
| *Mars* | jgm85f01 | 2002 | 85 | |
| *Mars* | ggm2bc80 | 2000 | 80 | |
| *Mars* | ggm50a01 | 1998 | 50 | |
| *Mars* | ggm50a02 | 1998 | 50 | |
| *Mars* | jgm50c01 | 1998 | 50 | |
| *Moon* | GrazLGM420b | 2018 | 420 | **Wirnsberger H. et al, 2018** |
| *Moon* | GrazLGM420b+ | 2018 | 420 | **Wirnsberger H. et al, 2018** |
| *Moon* | RFM_Moon_2520 | 2018 | 2520 | **Sprlak et al., (2018)** |
| *Moon* | GrazLGM420a | 2017 | 420 | **Wirnsberger H. et al, 2017** |
| *Moon* | dV_MoonTopo_2160 | 2017 | 2160 | **Hirt, C. and M. Kuhn (2017)** |
| *Moon* | GrazLGM300c | 2016 | 300 | **Krauss, S. et al, 2016** |
| *Moon* | AIUB-GRL200A | 2015 | 200 | **Arnold D. et al, 2015** |
| *Moon* | AIUB-GRL200B | 2015 | 200 | **Arnold D. et al, 2015** |
| *Moon* | GL0660B | 2013 | 660 | |
| *Moon* | GRGM660PRIM | 2013 | 660 | |
| *Moon* | JGL150Q1 | 2000 | 150 | |
| *Moon* | JGL165P1 | 2000 | 165 | |
| *Moon* | JGL100J1 | 1999 | 100 | |
| *Moon* | JGL100K1 | 1999 | 100 | |
| *Moon* | JGL075D1 | 1998 | 75 | |
| *Moon* | JGL075G1 | 1998 | 75 | |
| *Moon* | GLGM-2 | 1995 | 70 | **F. G. Lemoine et al, 1995** |
| *Moon* | GLGM-1 | 1994 | 70 | **F. G. Lemoine et al, 1994** |
| *Venus* | shgj180ua01 | 1997 | 180 | |
| *Venus* | shgj120pa01 | 1996 | 120 | |

