# Peer review of "ICGEM – 15 years of successful collection and distribution of global gravitational models, associated services and future plans"

_Earth System Science Data, 2019_

## Referee Comment (RC1) · Anonymous Referee #1 · 4 Mar 2019

General Comment

The manuscript provides a summary about the ICGEM service and its role in geodesy and other Earth sciences. I highly appreciate the preparation of such a manuscript in order to enhance the visibility of this service, which in my view is one of the best in geodesy. Indeed ICGEM nowadays in the central node for global gravity field models of various kind. The paper is very well written and I have no general comments related to structure and content. Nevertheless, in the following I provide some detailed comments which could be considered for the final revision. In summary I recommend a minor revision without the need for a re-review.

[Figure]

Detailed Comments

1. Introduction

The list of references in this section seems to be a little bit arbitrary. There are many other publications addressing the various items, which sometimes would be even better suited as they would be more of a review type. I would recommend to go through the list again and may be to cite mostly review papers. As a minimum I would add "e.g." in order to indicate that these are just examples of papers from a larger selection.

Page 2, line 21 to 32: I would recommend to mention the gravity gradients and measurements of non-gravitational accelerations right before mentioning the satellite missions (i.e. at line 24). Otherwise it sounds like these measurements are not taken from the satellites. Then one could continue after the satellites with: "Other fundamental datasets . . . . . . . . . are terrestrial gravity measurements from moving platforms . . . and collected on the Earth surface.

Page 4, line 9: I think the references to Drewes and Toth are not required here.

2. Background of the ICGEM Service

2.1 History of ICGEM

Page 5, Figure 2, Caption: Please make clear that compared to the satellite models the EIGEN6-C4 is based also on terrestrial data. May be re-phrase the following sentence: ". . . .. Note that the EIGEN6-C4 is not the truth but a better approximation to the real gravity field, because it includes terrestrial and altimetry derived gravity field information". Last sentence: I think it is EGM96S (and not EGM96).

Page 6, Line 1: Currently? Please indicate a date because numbers are changing .

Page 6, Line 10: Delete last part of the sentence: ". . . and promises future developments".

2.2 Scientific Background and ICGEM's Data

In general I think this section should be shortened and reference to the ICGEM documentation should be made. It is impossible to completely write the geoedetic gravity theory with all details in such a paper. I think the main purpose should be an explanation that global models are represented by spherical harmonics.

Page 9, Line 6-7: Update sentence: ". . ... pure gravitational forces . . ...the Earth's gravitational attraction (V) . . .. and potential of the centrifugal force due to Earth rotation".

Page 9, Line 21 to Page 10, Line 7: I think this is not needed here (including equ. (19 to (3). Instead reference to the ICGEM documentation or another book shall be made.

Page 11, Line 12: Find a better wording. Proposal: ". . .. normalisation is defined such that the average square . . .".

Page 13, Line 7-8: The geoid is introduced. In my opinion the definition is a bit misleading. Undisturbed in my view doesn't include the MDT because it is a permanent disturbance and therefore the statement is not correct. I would write as follows: The geoid is an equipotential surface that in average approximates the mean sea surface.

Page 13, Equation (10): I would recommend to write equation (7) and (10) in the same way.

Page 13, Line 24: Why can these quantities be computed approximately? The calculation is correct, just the models are incomplete. Gravity disturbances can be computed exactly, while for geoid oen needs to do assumptions. Please re-phrase.

Page 14, Line 5: "above the geoid" instead "over"

Page 14, Equations (11) and (12): I don't think this is needed here. Just refer to the manual. Otherwise one should provide equations for all derived quantities.

2.2.1 Static global gravity field models of the Earth

Page 15, Line 4: EGM2008 expansion to degree and order 2159 is for ellipsoidal harmonics. After conversion to spherical harmonics the expansion is up to degree 2190.

Modify the sentence accordingly.

Page 15, Line 9: Write: "As an example one of the high resolution . . .."

Page 15, Line 18: Is the gravity attraction really stronger? This depends where you observe it. If you stay on the equipotential surface the gravity attraction becomes smaller because the neighbouring equipotential surfaces are separated by a larger distance. Please rethink this sentence and be more specific.

Page 15, Lines 19-24: This is confusing and in my view, specifically the sentence about the North Atlantic (why only there?). Please rephrase and leave out unclear statements.

Page 17, Line 5: "over land" not "in the land".

2.2.2 Temporal global gravity field models

Page 19, Line 4: I think monthly gravity field models do not provide a resolution of 160 km. May be only when looking to the maximum degree of the SHS but not in the sense of real data content. Please make this clear.

Page 19, Line 16: It is not only water mass, but could also be geophysical signals (solid Earth).

Page 20, Line 9: Reference Wahr, 2007 is missing.

2.2.3 Topographic global gravity field models

Page 21, Figure 10, Caption: a) and b) is not indicated in the sub-plots. Instead write left and right.

Page 21, Table 1: Monitoring sea level variations is a temporal gravity field signal if the pure mass variation is meant. If just the geometric change of the sea level is meant it is no gravity field signal at all. Please correct. Correct also the Atmosphere section (no underlining).

**3. Services of ICGEM**

**3.1 Calculation Service**

Page 23, Line 7-8: I think the semi major axis is missing here.

Page 23, Lines 19-27: This section is quite confusing and I would recommend to rephrase it in simpler words as it is not very clear to non-experts. May be it would be again sufficient to refer to the ICGEM manual.

Page 25, Table 2: For second_r_derivative one could also write vertical gravity gradient. This is a more convenient name. Just a proposal for future development: Why not offering also th3e horizontal gravity gradients. These might be useful for some purposes.

**3.2 Visualisation Service**

Page 31, Figure 16, Caption: For b) I think it should be written ". . .. Represent the mass change." Instead of "distribution".

**3.3 Evaluation of global gravity field models**

**3.3.1 Model evaluation with respect to other models in the spectral domain**

Page 32, bottom: wrong font

Page 33, Figure 17: I think the green line are the "Cumulative difference amplitudes . . ..". Please correct

**3.3.2 Model evaluation with respect to GNSS/levelling derived geoid undulations**

In my view this chapter either needs to be significantly extended or its value is very limited as the procedure to do comparisons with GPS/levelling geoid heights is much more complicated as it is done here. Therefore the numbers provided in figure 18 are not really meaningful, e.g. the omission error is not considered at all. In the last paragraph of page 34 the authors even explain that this is not a fair comparison. So why

do they show it or why is it offered in ICGEM at all? There are also missing references to publications dealing with GPS/levelling comparisons. I would consider to delete the complete paragraph and even to consider not to offer this in ICGEM as long as it is not a fair comparison. It only provides misleading results to the not experienced user.

Appendix 2:

I am not sure if this is really needed. Why not setting a link to the web site with the models.

---

## Referee Comment (RC2) · Anonymous Referee #2 · 8 Mar 2019

General Comments The manuscript provides a description of ICGEM, being IAG/IGFS's unique and worldwide greatly appreciated user service for the collection, archiving, DOI assignment and dissemination of global gravity field models, determined from a variety of satellite and terrestrial observables by various analysis approaches, a service which in addition to these basic service functions provides also online calculation and visualization tools to relieve the user of the task of accurately calculating and visualizing various gravity functionals on selected grids by himself. The paper, providing up to date material and covering all main aspects for the use of the ICGEM service, of relevance for the readership of the journal, is well structured into seven sections of quite different length and language precision (examples are given below). Over all, in

the reviewer's opinion the paper is well written, but is too long and needs some revision. Because of the importance of this unique gravity product service for research activities in the geosciences he recommends that a revised version be published after a re-review. Specific comments Abstract The manuscript title suggests that in addition to information on models and services presented in sections 2 and 3, something will be said about future plans. Scattered information on future plans can be found in subsection 2.1 History and 6 Conclusions, but wouldn't it be better to introduce a separate section on Future Plans or to rename subsection 2.1 into History, Status and Future Planning? P.1, L.17/18: modify to "including those from the 1960s to the 1990s, as well as the most recent ones" P.1, L.19: "such as satellite" should read "such as satellite altimetry and…" P.1, L.23/24: polish text P.1, L. 25: paper does not present models. Change to ..We present a list of static, temporal…

1. Introduction

Introduction is too long, needs shortening P.2, L.5: change to "..in the 1960s to 1990s" P.2, L.6: Better "mass change" P.2, L.17: I suspect "analysis techniques" is meant P.2, L.18,20 Drewes et al., 2016, why the 3 references in Line 20? P.2, L.22: satellite orbit perturbations are derived quantities, not satellite observations P.2, L.30: imprecise phrasing: "terrestrial" gravity measurements collected on the "Earth surface" P.3, L.5: imprecise phrasing: new measurements become available from…terrestrial measurements P.3, L.7: modify wording- in reality ICGEM is not a meeting point, but acts as an interface P.3, L.11-L.18. For this paper, the text parts on IAG and IGFS are not of particular relevance. In the opinion of the reviewer it is sufficient to leave the text with the URLs in lines 9 and 10, complemented perhaps by the reference Drewes et al., 2016, and to delete the text part up to line 18. The same applies to page 4 line 17 for the four IGFS services and delete lines 18-26. P.4, L.9: why these references? P.4, L.31: modify text into "various types of gravity field models P.5, L.3: and section 7?

2. The Background of the ICGEM Service

2.1 History P.5, L.6: change perhaps to History, Status and Future Planning of ICGEM. P.5, L8: reference to the 1997 IAGA resolution No 1 for an international Decade of Geopotential Missions would be useful at this point.http://www.iaga-aiga.org/index.php?id=res1-97 P.5, L.19: add developed "by various institutions" P.5, L.20: add "dedicated" gravity missions P.6, L.7: add some wording that EIGEN-6C4 is a combination solution derived from. . ..

2.2 Scientific background and ICGEM's Data P.9, L.1: misleading wording: ICGEM does not provide data but gravity model related quantities. P.9, L.7: polish wording "including on the ground. . .." P.9, L.21 to P.14, L.10: Basic equations of the potential of a solid body, of the spherical harmonics expansion of the gravitational potential and the disturbing potential can be found in many textbooks and this text part should considerably be shortened and reformulated. Eq. 7 should be sufficient to explain the content of the ICGEM product archive and the characteristics of the expansion into spherical harmonics. For the various functionals reference could be made to Barthelmes 2013.

2.2.1 Static gravity field models P.14, L.5: replace by "above geoid " P.15, L.22: sloppy formulation . . ..underneath the Earth! P.15, L.25-30: What's the point at this point of the text? Wouldn't it better fit to future aspects?

2.2.2 Temporal global gravity field models P.19, L.2: improve wording:. . ..as for longer P.19, L.4: change to 300 km

2.2.3 Topographic global gravity field models P.21, L.1: designation missing for plots P.21, L.7: in table 1- for sea level, ice and atmosphere only mass change can be meant.

2.2.4 Models of other celestial bodies P.22, L.7: polish wording P.22, L.9: add "the presently most detailed (or best) gravitational field"

3. Service of ICGEM

3.1 Calculation Service P.22, L.13: change have to has

3.2 3D Visualization Service P.29, L.27: improve wording by beginning "clearly visible

in these representations is. . ." P.31, L.24: mass change is more appropriate

3.3 Evaluation of global gravity field models P.32, L.14-L.21: different fond P.34, L.25ff: what is the purpose of the quick check assessments of the service, if not allowing fair comparisons?

3.5 DOI Service P.36, L.27: polish text " and is equipped " ??

4. Documentation P.38, L.14: polish wording "scientific disciplines and industry related background" ?? P.39, L.4: change to "pays "

6. Conclusions and future aspects P.42, L. 6 ff: see remark under Abstract

––––––––––––––––––––––––––––

---

## Referee Comment (RC3) · Anonymous Referee #3 · 8 Mar 2019

The paper gives an extensive description of the scientific activities of ICGEM. The paper is unusually long but I think that its length is adequate to fully and properly describe the services provided by ICGEM. Also, the paper is well organised and written in a good English language. I only have some minor comments on the paper that are listed below.

- page 9, 15: the discussion on the terms "gravitational" and "gravity" is quite misleading. I don't agree with the authors' statement, i.e. to use "gravity" instead of "gravitational". I think that we must stay strictly in the geodetic tradition and use properly the two terms throughout the paper.

[Figure]

- page 10, 10-15: this comment connects to the previous one. The authors stated that "gravity" has to be used and then they write "Geodesy describes the gravitational potential only in empty space,...". This is not in the line that they stated. So, again, I would ask the authors to stick to the standard notation of Geodesy, which is clear, without contradiction and used for many years.

- page 10: it is frequently used the sentence "real gravity field". I would use "gravity field" only

- page 11, Eq. (8): Pnm is normalised so it should have the bar on top.

- page 13, before Eq. (10): "and valid in space". I would write "in space"

- page 17, 10: "physical heights". I would add "physical heights (i.e. orthometric heights)"

- page 23, 20-25. I would skip the sentence "(ellipsoidal equipotential...over the oceans)" which could be misleading

- page 29, 15: instead of "different models quickly" write "different models" because "quickly" is written in the same line.

- page 32, Eq. (13): please replace "s" withe the greek letter sigma to be coherent with the statement above

---

## Author Comment (AC1) · 19 Mar 2019

We would like to thank the reviewer for his time and constructive feedback. We believe the current content of the paper is now clarified from the point of few items the reviewer suggested. Below we added our responses to reviewer's detailed comments in bold.

*The paper gives an extensive description of the scientific activities of ICGEM. The paper is unusually long but I think that its length is adequate to fully and properly describe the services provided by ICGEM. Also, the paper is well organised and written in a good English language. I only have some minor comments on the paper that are listed below.*

**In the revised version, we tried to reduce the size of the paper in some sections as recommended by the other two reviewers to focus on the more important items. To keep the size of the entire paper smaller, we also removed some of the Appendices that included the list of the models. Considering the size of the figures in general, we hope the professional editing from the journal would help the readers to follow the paper easily.**

*- page 9, 15: the discussion on the terms "gravitational" and "gravity" is quite misleading. I don't agree with the authors' statement, i.e. to use "gravity" instead of "gravitational". I think that we must stay strictly in the geodetic tradition and use properly the two terms throughout the paper.*

**This topic on the terminology has been discussed also among the authors and indeed we agree that we should stick with the correct terminology and be consistent. For the purpose of this paper, the clarification was clearly made from the point of the difference between the two to prevent any confusion. The terminology used through paper that is related to the coefficients of the models provided would refer to "gravitational". For the rest of the paper, we paid more attention to be consistent and correct with the use of the "gravity" and "gravitational". Thank you for noting this very important item.**

*- page 10, 10-15: this comment connects to the previous one. The authors stated that "gravity" has to be used and then they write "Geodesy describes the gravitational potential only in empty space,...". This is not in the line that they stated. So, again, I would ask the authors to stick to the standard notation of Geodesy, which is clear, without contradiction and used for many years.*

**In the revised version, we have changed at few places gravity into gravitational for the correct use of the terminology and consistency. Thank you for paying attention to this point.**

*- page 10: it is frequently used the sentence "real gravity field". I would use "gravity field" only*

**This is to prevent the confusion between the model approximations to the gravity field and the actual gravity field itself. One can use "true gravity field" instead as well. Since gravity field is repeated many times, we would like to distinguish them by using "real".**

*- page 11, Eq. (8): Pnm is normalised so it should have the bar on top.*

**Added.**

*- page 13, before Eq. (10): "and valid in space". I would write "in space"*

**Replaced.**

*- page 17, 10: "physical heights". I would add "physical heights (i.e. orthometric heights)"*

**Added as suggested.**

*-page 23, 20-25. I would skip the sentence "(ellipsoidal equipotential...over the oceans)" which could be misleading*

**The sentence starting with "The defining parameters.." has been removed for clarity.**

*- page 29, 15: instead of "different models quickly" write "different models" because "quickly" is written in the same line.*

**Indeed, it is edited in the revised version.**

*- page 32, Eq. (13): please replace "s" withe the greek letter sigma to be coherent with the statement above*

**This was due to the conversion to pdf, in the revised version it is paid attention to represent it correctly.**

---

## Author Comment (AC2) · 19 Mar 2019

We would like to thank the reviewer for accepting reviewing our paper and submitting his comments within the scheduled time. We appreciate the detailed comments and feedback with high level of expertise which very much helped to improve the paper. In general, we agree with most of the comments and we will further apply or have already applied suggested changes in the revised version. Also, there were parts in the paper where we were uncertain about some ideas, such as whether to add the list of the models in the Appendix and the reviewer just naturally provided us feedback on such items as well. We would like to thank him for his time and respond below to his detailed comments. The reviewer's comments are indicated in italic fonts whereas our responses are given in bold.

*Detailed Comments*

*1. Introduction*

*The list of references in this section seems to be a little bit arbitrary. There are many other publications addressing the various items, which sometimes would be even better suited as they would be more of a review type. I would recommend to go through the list again and may be to cite mostly review papers. As a minimum I would add "e.g." in order to indicate that these are just examples of papers from a larger selection.*

**In general we tried to refer to the most fundamental (e.g. first of its kind, most cited) publications in the relevant topics but I could see this can be better linked in this section. We scanned through the references carefully and removed some of the references and introduced others that are more of review kind. The revised version should be more consistent in terms of the reference list.**

*Page 2, line 21 to 32: I would recommend to mention the gravity gradients and measurements of non-gravitational accelerations right before mentioning the satellite missions (i.e. at line 24). Otherwise it sounds like these measurements are not taken from the satellites. Then one could continue after the satellites with: "Other fundamental datasets . . .. . .. . . are terrestrial gravity measurements from moving platforms . . . and collected on the Earth surface.*

**We have moved the "gravity gradients and non-gravitational accelerations" to the previous sentence and modified as recommended.**

*Page 4, line 9: I think the references to Drewes and Toth are not required here.*

**The content of Drewes and Toth are from ICGEM and we thought they would be good links to the available documents; but they could be removed as well which we did in the revised version.**

2. Background of the ICGEM Service

2.1 History of ICGEM

*Page 5, Figure 2, Caption: Please make clear that compared to the satellite models the EIGEN6-C4 is based also on terrestrial data. May be re-phrase the following sentence: ". . .. Note that the EIGEN6-C4 is not the truth but a better approximation to the real gravity field, because it includes terrestrial and altimetry derived gravity field information". Last sentence: I think it is EGM96S (and not EGM96).*

**We have revised the caption indicating clearly the content of the EIGEN-6C4. Thank you for your suggestion, indeed the model is EGM96S and not EGM96 which is also modified in the caption.**

*Page 6, Line 1: Currently? Please indicate a date because numbers are changing .*

**"By January 2019" is added.**

*Page 6, Line 10: Delete last part of the sentence: ". . . and promises future developments".*

***Removed.***

*2.2 Scientific Background and ICGEM's Data*

*In general I think this section should be shortened and reference to the ICGEM documentation should be made. It is impossible to completely write the geoedetic gravity theory with all details in such a paper. I think the main purpose should be an explanation that global models are represented by spherical harmonics.*

**This was a previous discussion we eventually had while writing this section. We agree that the potential theory cannot be given in such a paper as complete. However, while writing the paper, we thought we would like to provide at least the equations which can explain the link between the global models and spherical harmonics. In order to provide a smooth introduction and transition later, we thought including the basics of the Newton's law of gravitation would help to explain the potential theory. Thank you for commenting on this. Not to lose the focus of the paper and to make it more readable, the revised version has this part reduced. We only kept a few formulations representing the widely used functionals and their representation in terms of spherical harmonic coefficients.**

*Page 9, Line 6-7: Update sentence: ". . . .. pure gravitational forces . . ...the Earth's gravitational attraction (V) . . .. and potential of the centrifugal force due to Earth rotation".*

**Updated.**

*Page 9, Line 21 to Page 10, Line 7: I think this is not needed here (including equ. (19 to (3). Instead reference to the ICGEM documentation or another book shall be made.*

**We removed the first 3 equations and referred to the Advanced Physical Geodesy Text book and Scientific report from Barthelmes, 2013.**

*Page 11, Line 12: Find a better wording. Proposal: ". . .. normalisation is defined such that the average square . . .".*

**Applied.**

*Page 13, Line 7-8: The geoid is introduced. In my opinion the definition is a bit misleading. Undisturbed in my view doesn't include the MDT because it is a permanent disturbance and therefore the statement is not correct. I would write as follows: The geoid is an equipotential surface that in average approximates the mean sea surface.*

**The formulation of geoid is kept simple as recommended and only the equipotential surface is mentioned in the revised version. References are provided for the interested readers.**

*Page 13, Equation (10): I would recommend to write equation (7) and (10) in the same way.*

**Written as recommended.**

*Page 13, Line 24: Why can these quantities be computed approximately? The calculation is correct, just the models are incomplete. Gravity disturbances can be computed exactly, while for geoid oen needs to do assumptions. Please re-phrase.*

**It is true that the gravity disturbance can be computed exactly. However, here in the paper we introduced the disturbing potential and summarized that the disturbing potential is used in the calculation of some of the functionals such as geoid which can be done only via approximations (especially for the grid calculations). Gravity disturbance however can be computed from W and U (not T), exactly from spherical harmonic coefficients with no approximation introduced. We clarified this in the text to avoid confusion.**

*Page 14, Line 5: "above the geoid" instead "over"*

**Indeed, applied as suggested.**

*Page 14, Equations (11) and (12): I don't think this is needed here. Just refer to the manual. Otherwise one should provide equations for all derived quantities.*

**We think having these two equations used in the calculation of two main functionals may provide good references for the rest. How the coefficients are included in the computations may be summarized with these two.**

*2.2.1 Static global gravity field models of the Earth*

*Page 15, Line 4: EGM2008 expansion to degree and order 2159 is for ellipsoidal harmonics. After conversion to spherical harmonics the expansion is up to degree 2190. Modify the sentence accordingly.*

**Thank you for reminding. Corrected.**

*Page 15, Line 9: Write: "As an example one of the high resolution . . .."*

**Modified.**

*Page 15, Line 18: Is the gravity attraction really stronger? This depends where you observe it. If you stay on the equipotential surface the gravity attraction becomes smaller because the neighbouring equipotential surfaces are separated by a larger distance. Please rethink this sentence and be more specific.*

**Maybe our explanation was not clear. The point of observation has to stay exactly the same, which would not be possible anymore under the circumstances introduced (e.g. switch on). Explanation of the features is quite sophisticated as one can imagine; therefore, we decided to remove the above mentioned part and simplify this paragraph, and refer to the dedicated study about the Indian Ocean geoid instead.**

*Page 15, Lines 19-24: This is confusing and in my view, specifically the sentence about the North Atlantic (why only there?). Please rephrase and leave out unclear statements.*

**This part has been removed from the text. The same principle applied to other areas as well, but North Atlantic was chosen as an example.**

*Page 17, Line 5: "over land" not "in the land".*

**Applied.**

*2.2.2 Temporal global gravity field models*

*Page 19, Line 4: I think monthly gravity field models do not provide a resolution of 160 km. May be only when looking to the maximum degree of the SHS but not in the sense of real data content. Please make this clear.*

**Indeed ~160 km would only be possible with the highest degree/order available which is in fact not the case in practice due to the increasing error of the higher degree/order coefficients. Therefore, this part has been revised and ~300 km for monthly solutions is used instead.**

*Page 19, Line 16: It is not only water mass, but could also be geophysical signals (solid Earth).*

**The geophysical signals added in the sentence.**

*Page 20, Line 9: Reference Wahr, 2007 is missing.*

**Thank you for noting this; the reference is added to the list.**

*2.2.3 Topographic global gravity field models*

*Page 21, Figure 10, Caption: a) and b) is not indicated in the sub-plots. Instead write left and right.*

**Letters are brought to the front, now they are visible.**

*Page 21, Table 1: Monitoring sea level variations is a temporal gravity field signal if the pure mass variation is meant. If just the geometric change of the sea level is meant it is no gravity field signal at all. Please correct.*

*Correct also the Atmosphere section (no underlining).*

**We have replaced "level" with "mass". We don't mean to go into details with steric and non-steric sea level changes. Underline in Atmosphere is also removed.**

*3. Services of ICGEM*

*3.1 Calculation Service*

*Page 23, Line 7-8: I think the semi major axis is missing here.*

**Thank you for noticing and noting this. Radius corresponds to the semi-major axis here, we added this information within parenthesis in the text. We will eventually need to replace it in the service as well since this may cause confusion.**

*Page 23, Lines 19-27: This section is quite confusing and I would recommend to rephrase it in simpler words as it is not very clear to non-experts. May be it would be again sufficient to refer to the ICGEM manual.*

**Some reductions are applied for clarity purposes.**

*Page 25, Table 2: For second_r_derivative one could also write vertical gravity gradient. This is a more convenient name.*

**Added in parenthesis. However, in fact it is not the same for the computations introduced in the ICGEM Calculation Service. "Vertical" (plumb line approx. by normal direction) is not radial (spherical approx.). Therefore, second_r_ direction is an approximation of vertical gravity gradient.**

*Just a proposal for future development: Why not offering also the horizontal gravity gradients. These might be useful for some purposes.*

**This has been in our to-do-list. We also added this to the future work in the paper. However, we suspect that providing this in ICGEM is not as easy as it sounds. Some standards (e.g. definition of horizontal, which coordinate system to be used) need to be clarified from the users point of view. The authors would be happy to hear about what would be interesting for the community.**

*3.2 Visualisation Service*

*Page 31, Figure 16, Caption: For b) I think it should be written ". . .. Represent the mass change." Instead of "distribution".*

**Applied.**

*3.3 Evaluation of global gravity field models*

*3.3.1 Model evaluation with respect to other models in the spectral domain*

*Page 32, bottom: wrong font*

**Replaced with the correct font size.**

*Page 33, Figure 17: I think the green line are the "Cumulative difference amplitudes . . ..". Please correct*

**This terminology was recommended in the past, but for the geodetic community indeed the cumulative would be more familiar. In the revised version, it is edited as recommended.**

*3.3.2 Model evaluation with respect to GNSS/levelling derived geoid undulations*

*In my view this chapter either needs to be significantly extended or its value is very limited as the procedure to do comparisons with GPS/levelling geoid heights is much more complicated as it is done here. Therefore the numbers provided in figure 18 are not really meaningful, e.g. the omission error is not considered at all. In the last paragraph of page 34 the authors even explain that this is not a fair comparison. So why do they show it or why is it offered in ICGEM at all? There are also missing references to publications dealing with GPS/levelling comparisons. I would consider to delete the complete paragraph and even to consider not to offer this in ICGEM as long as it is not a fair comparison. It only provides misleading results to the not experienced user.*

**We agree with reviewer's concern on the contribution of GNSS/levelling evaluation to the use of ICGEM Service. Taking into account the omission error, this method can for sure be applied in more sophisticated ways. However, in general, ICGEM would like to compare the models among themselves (e.g. among satellite-only models, among combined models) and does not seek for absolute comparisons for particular degrees. Therefore, we think this kind of comparisons still provide useful information of different models wrt the same external datasets. We try to make use of the GNSS/levelling series we have been delivered in the past which was scanned for outliers initially. We think the evaluation provided in this section is still valuable and should be kept in the ICGEM which may be improved in the future. References concerning the GNSS/leveling evaluation are added in the revised version.**

*Appendix 2: I am not sure if this is really needed. Why not setting a link to the web site with the models.*

**This is something we could not decide in the beginning. In the revised version we removed the list of the models.**

---

## Author Comment (AC3) · 19 Mar 2019

**We would like to thank the reviewer for his comments that have helped to improve the readability of the paper and increase the precision of the language and the statements. We respond to each comment in detail below as indicated in bold. Reviewer's original comments can be found in italic font.**

*Detailed Comments*

*Abstract*

*The manuscript title suggests that in addition to information on models and services presented in sections 2 and 3, something will be said about future plans. Scattered information on future plans can be found in sub- section 2.1 History and 6 Conclusions, but wouldn't it be better to introduce a separate section on Future Plans or to rename subsection 2.1 into History, Status and Future Planning?*

**Initially, we wanted to introduce the service, and provide information about its components through the paper. Then we thought collecting all the future plans at the end of the paper would be ideal and conclude the paper. Nevertheless, presenting them in Section 2.1 might be a good idea as well. In the revised version, it is applied as suggested by the reviewer.**

*P.1, L.17/18: modify to "including those from the 1960s to the 1990s, as well as the most recent ones"*

**Applied as suggested by the reviewer.**

*P.1, L.19: "such as satellite" should read "such as satellite altimetry and. . ."*

**Added.**

*P.1, L.23/24: polish text*

**This sentence has been removed from the abstract and included in the rest of the manuscript in different forms.**

*P.1, L. 25: paper does not present models. Change to ..We present a list of static, temporal. . .*

**The sentence has been rephrased to "We present the ICGEM's data by means of ….."**

1. Introduction

*Introduction is too long, needs shortening*

**Some reductions are applied also to the introduction (e.g. removing Fig. 1), besides data and other sections.**

*P.2, L.5: change to "..in the 1960s to 1990s"*

**1990s is added..**

*P.2, L.6: Better "mass change"*

**Edited**.

*P.2, L.17: I suspect "analysis techniques" is meant*

**We have replaced with the technologies.**

*P.2, L.18,20 Drewes et al., 2016, why the 3 references in Line 20?*

**Drewes and Toth include actually the ICGEM reports for different periods which include the mentioned information in the text. But, to avoid confusion and make the text more readable the references are removed.**

*P.2, L.22: satellite orbit perturbations are derived quantities, not satellite observations*

**It is mentioned in the text as "…derived from GNSS measurements". Therefore, we agreed that they are not direct observations.**

*P.2, L.30: imprecise phrasing: "terrestrial" gravity measurements collected on the "Earth surface"*

**These refer to the gravity measurements collected on the Earth surface. For the purpose of this paper, the details given in this part would be enough from our point of view. But, any suggestion to make it more precise is very welcome.**

*P.3, L.5: imprecise phrasing: new measurements become available from. . .terrestrial measurements*

**Since inclusion of such information would further increase the size of the paper, we have avoided adding more information on such items.**

*P.3, L.7: modify wording in reality ICGEM is not a meeting point, but acts as an interface*

**We changed as the meeting platform which may replace the interface.**

*P.3, L.11-L.18. For this paper, the text parts on IAG and IGFS are not of particular relevance. In the opinion of the reviewer it is sufficient to leave the text with the URLs in lines 9 and 10, complemented perhaps by the reference Drewes et al., 2016, and to delete the text part up to line 18. The same applies to page 4 line 17 for the four IGFS services and delete lines 18-26. P.4,*

*L.9: why these references?*

**This is also a part where we discussed among the co-authors while writing the paper. We believe adding this part makes the position of such a service clear and complete the content of the paper. The need for the ICGEM Service is made obvious with the information and support provided by the association. Therefore, we would like to keep this part in the paper. However, to save some space, we removed Fig.1 and some more text.**

*P.4, L.31: modify text into "various types of gravity field models*

**Modified.**

*P.5, L.3: and section 7?*

**Added.**

*2. The Background of the ICGEM Service*

2.1 History

*P.5, L.6: change perhaps to History, Status and Future Planning of ICGEM.*

**Modified as suggested, and we moved some sentences from the Conclusion and Future Plans to this section.**

*P.5, L8: reference to the 1997 IAGA resolution No 1 for an interna tional Decade of Geopotential Missions would be useful at this point.http://www.iaga- aiga.org/index.php?id=res1-97*

**Added.**

*P.5, L.19: add developed "by various institutions"*

**Added.**

*P.5, L.20: add "dedicated" gravity missions*

**Added.**

*P.6, L.7: add some wording that EIGEN-6C4 is a combination solution derived from. . ..*

**Added in the caption.**

*2.2 Scientific background and ICGEM's Data*

*P.9, L.1: misleading wording: ICGEM does not provide data but gravity model related quantities.*

**ICGEM collects and distributes global gravity field models, as well as provides products via calculation service. Within this concept, after some discussions with data service in our institution, our understanding is any of the above mentioned can be called data in general. Therefore, the subtitle includes "Data" to refer to all these in general.**

*P.9, L.7: polish wording "including on the ground. . .."*

**This part has been removed in the revised version.**

*P.9, L.21 to P.14, L.10: Basic equations of the potential of a solid body, of the spherical harmonics expansion of the gravitational potential and the disturbing potential can be found in many textbooks and this text part should considerably be shortened and reformulated. Eq. 7 should be sufficient to explain the content of the ICGEM product archive and the characteristics of the expansion into spherical harmonics. For the various functionals reference could be made to Barthelmes 2013.*

**This part has been reduced as well in the revised version. We still think having the equations for the geoid undulation and gravity disturbances can be good references for the other functionals without the need of going to the references in the first place. For detailed information, the complete references either Barthelmes 2013 and other textbooks are given.**

*2.2.1 Static gravity field models*

*P.14, L.5: replace by "above geoid "*

**Indeed. Replaced in the revised version.**

*P.15, L.22: sloppy formulation . . ..underneath the Earth!*

**Replaced with "in the deep mantle" as given in the original reference.**

*P.15, L.25-30: What's the point at this point of the text? Wouldn't it better fit to future aspects?*

**It is a link to the preliminary model of EGM2020, XGM2016 which is shortly covered in one of the examples provided in the paper. Moreover, since these are future aspect of the global gravity field models and not the ICGEM Service directly, it may be more suitable to keep it in this section.**

*2.2.2 Temporal global gravity field models*

*P.19, L.2: improve wording:. . ..as for longer*

**Edited.**

*P.19, L.4: change to 300 km*

**This was written considering the maximum degree and order expansion of the temporal models. In the revised version, we have replaced it with ~300 km for monthly solutions.**

*2.2.3 Topographic global gravity field models*

*P.21, L.1: designation missing for plots*

**Letters are added to the figures.**

*P.21, L.7: in table 1- for sea level, ice and atmosphere only mass change can be meant.*

**A note to the caption is added for clarity "Note that the variations refer to the mass change".**

*2.2.4 Models of other celestial bodies*

*P.22, L.7: polish wording*

**Original sentence" These models are also developed based on similar observations of the gravity field of the body.**

Replacement: **These models are also developed based on similar observations.**

*P.22, L.9: add "the presently most detailed (or best) gravitational field"*

**Revised version:" ….have been used to develop the most detailed gravitational field of the Moon so far".**

*3. Service of ICGEM*

*3.1 Calculation Service P.22, L.13: change have to has*

**Indeed. Replaced.**

*3.2 3D Visualization Service*

*P.29, L.27: improve wording by beginning "clearly visible in these representations is. . ."*

**Applied.**

*P.31, L.24: mass change is more appropriate*

**Indeed, applied as suggested.**

*3.3 Evaluation of global gravity field models*

*P.32, L.14-L.21: different fond*

**Edited.**

*P.34, L.25ff: what is the purpose of the quick check assessments of the service, if not allowing fair comparisons?*

**Maybe we should have used another wording instead of "fair". The service still performs very useful comparisons among the models but does not apply sophisticated comparisons that are interest of many other studies published individually. Since the ICGEM's aim is to compare various models wrt exactly the same external, independent datasets, the results still serve for basic comparison purposes. In the revised version more explanations and future plans for the GNSS/levelling comparisons are added.**

*3.5 DOI Service*

*P.36, L.27: polish text " and is equipped " ??*

**We have changed the original text "and is equipped with" to "Metadata can be harvested via an Application Programing Interface (OAI-PMH)."**

*4. Documentation*

*P.38, L.14: polish wording "scientific disciplines and industry related background" ??*

**Industry part is removed in the revised version.**

*P.39, L.4: change to "pays "*

**Indeed, thank you for noting.**

*6. Conclusions and future aspects*

*P.42, L. 6 ff: see remark under Abstract*

**Applied as recommended in Section 2.1**

---

## Referee Comment (RC4) · Anonymous Referee #1 · 20 Mar 2019

Thanks a lot to the authors for updating and explaining the various items I adressed in my initial review. I think the paper has been improved and can be accepted in the present form for publication in ESSD.
* * *

---

## Short Comment (SC1) · 20 Mar 2019

[revised manuscript text omitted]

---

## Author Comment (AC4) · 20 Mar 2019

[revised manuscript text omitted]

---

## Referee Comment (RC5) · Anonymous Referee #2 · 21 Mar 2019

I enjoyed reading the revised, now more compact version of the manuscript. After a minor last correction on page 20, line 8, I think the revised paper is ready for publication in essd
* * *

---

## Referee Report (RR1)

The manuscript clearly improved after the authors had taken into account the overwhelming number of comments and corrections made by reviewers. The paper now also has a reasonable length, especially considering the authors' remark in the introduction that the individual sections are written independently of each other, so that the interested reader has to read only individual sections and not the whole paper.

The article provides readers with valuable background information, hints and examples for the use of the ICGEM services and is in the reviewers´ opinion ready for publication after the correction of the following minor misprints or omissions.

Page 2, Line 17: correct reference Barthelmes et al., 2016 on page 41, line 14 to 2017,

Page 2, Line 26: correct Poseison to Poseidon,

Page 9, Line 9: correct Heiskanen and Moritz, 1969 to 1967,

Page 10, Line 4: correct $S_{10}$ to $S_{11}$,

Page 10, Line 11: modify to develop a gravitational…,

Page 10, Line 12: delete 20000,

Page 13, Line 26: Förste et al., 2016b is a reference for EIGENS4 and should be deleted here,

Page 14, Line 10: interior. Full stop is missing,

Page 17, Line 14: Correct Bettadpur, 2012 and Watkins and Yuan, 2014 to 2012 (see reference),

Page 18, Line 28:  Fig. 9. Full stop is missing,

Page 20, Line 8: 3 Services of ICGEM should become should become new chapter heading in bold letters,

Page 26, Line 19: Fig. 14 should read Fig. 14a,

Page 26, Line 21: Fig. 15b should read Fig 14b,

Page 31, Line 12: insert ”is” provided,

Page 31, Line 25: correct Ince et al., 2011 to 2012 (see reference),

Page 36, Line 5: modify to gravitational models,

Page 39, Line 23: delete ”is” and add ” , “.